# Constructing an Optimal Behavior Basis for the Option Keyboard

**Lucas N. Alegre**
Institute of Informatics
Federal University of Rio Grande do Sul
Porto Alegre, RS, Brazil
lnalegre@inf.ufrgs.br

**Ana L. C. Bazzan**
Institute of Informatics
Federal University of Rio Grande do Sul
Porto Alegre, RS, Brazil
bazzan@inf.ufrgs.br

**André Barreto**
Google DeepMind
London, UK
andrebarreto@google.com

**Bruno C. da Silva**
University of Massachusetts
Amherst, MA, USA
bsilva@cs.umass.edu

## Abstract

Multi-task reinforcement learning aims to quickly identify solutions for new tasks with minimal or no additional interaction with the environment. Generalized Policy Improvement (GPI) addresses this by combining a set of base policies to produce a new one that is at least as good—though not necessarily optimal—as any individual base policy. Optimality can be ensured, particularly in the linear-reward case, via techniques that compute a Convex Coverage Set (CCS). However, these are computationally expensive and do not scale to complex domains. The Option Keyboard (OK) improves upon GPI by producing policies that are at least as good—and often better. It achieves this through a learned meta-policy that dynamically combines base policies. However, its performance critically depends on the choice of base policies. This raises a key question: is there an optimal set of base policies—an optimal *behavior basis*—that enables zero-shot identification of optimal solutions for *any* linear tasks? We solve this open problem by introducing a novel method that efficiently constructs such an optimal behavior basis. We show that it significantly reduces the number of base policies needed to ensure optimality in new tasks. We also prove that it is strictly more expressive than a CCS, enabling particular classes of *non-linear* tasks to be solved optimally. We empirically evaluate our technique in challenging domains and show that it outperforms state-of-the-art approaches, increasingly so as task complexity increases.

## 1 Introduction

Reinforcement learning (RL) methods have been successfully used to solve complex sequential decision-making problems (Silver et al., 2017; Bellemare et al., 2020). However, traditional RL algorithms typically require thousands or millions of interactions with the environment to learn a *single* policy for a *single* task. Multi-task RL methods address this limitation by enabling agents to quickly identify solutions for new tasks with minimal or no additional interactions. Such methods often achieve this by learning a *set* of specialized policies (or *behavior basis*) designed for specific tasks and subsequently combining them to more rapidly solve novel tasks.

A powerful approach for combining policies to solve new tasks in a zero-shot manner leverages *successor features* (SFs) and *generalized policy improvement* (Barreto et al., 2018, 2020). GPI can combine base policies to solve a new task, producing a policy that is at least as good as any individual base pol-

39th Conference on Neural Information Processing Systems (NeurIPS 2025).

icy. Importantly, however, GPI policies are not guaranteed to be optimal. Optimality can be ensured if tasks can be expressed as a linear combination of reward features using techniques that compute a *convex coverage set* (CCS) (Alegre et al., 2022). A CCS is a set of policies that includes an optimal policy for any linear task. Unfortunately, the number of policies in a CCS often grows exponentially with the number of reward features (Roijers, 2016), making existing algorithms computationally expensive and difficult to scale to complex domains (Yang et al., 2019; Alegre et al., 2022, 2023b).

The *option keyboard* (OK) method improves upon GPI by producing policies that are at least as good—and often better. It achieves this through a learned meta-policy that dynamically combines base policies by assigning state-dependent linear weights to reward features (Barreto et al., 2019). This allows OK to express a larger spectrum of behaviors than GPI and potentially solve tasks beyond linear rewards, enabling it to produce policies that GPI cannot represent. However, its performance critically depends on the choice of base policies available. Existing OK-based methods assume a predefined set of base policies or rely on expert knowledge to construct one, without addressing how to identify a good behavior basis (Barreto et al., 2019; Carvalho et al., 2023b). This raises a key question and open problem: is there an optimal set of base policies for the OK—an optimal *behavior basis*—that enables zero-shot identification of optimal solutions for *any* linear tasks?

We solve this open problem by introducing **O**ption **K**eyboard **B**asis (OKB), a novel method with strong formal guarantees that efficiently identifies an optimal behavior basis for the OK. OKB provably identifies a set of policies that allows the OK to optimally solve any linear task; i.e., it can express all policies in a CCS. We show that it significantly reduces the number of base policies required to ensure zero-shot optimality in new tasks. Furthermore, we prove that the set of policies it can express is strictly more expressive than a CCS, enabling particular classes of non-linear tasks to be solved optimally. We empirically evaluate our method in challenging high-dimensional RL problems and show that it consistently outperforms state-of-the-art GPI-based approaches. Importantly, we also observe that the performance gain over competing methods becomes more pronounced as the number of reward features increases.

## 2 Background

### 2.1 Reinforcement Learning

An RL problem (Sutton and Barto, 2018) is typically modeled as a *Markov decision process* (MDP). An MDP is defined as a tuple $M \triangleq (\mathcal{S}, \mathcal{A}, p, r, \mu, \gamma)$, where $\mathcal{S}$ is a state space, $\mathcal{A}$ is an action space, $p(\cdot|s, a)$ describes the distribution over next states given that the agent executed action $a$ in state $s$, $r : \mathcal{S} \times \mathcal{A} \times \mathcal{S} \to \mathbb{R}$ is a reward function, $\mu$ is an initial state distribution, and $\gamma \in [0, 1)$ is a discounting factor. Let $S_t$, $A_t$, and $R_t \triangleq r(S_t, A_t, S_{t+1})$ be random variables corresponding to the state, action, and reward, respectively, at time step $t$. The goal of an RL agent is to learn a policy $\pi : \mathcal{S} \to \mathcal{A}$ that maximizes the expected discounted sum of rewards (*return*), $G_t = \sum_{i=0}^{\infty} \gamma^i R_{t+i}$. The action-value function of a policy $\pi$ is defined as $q^\pi(s, a) \triangleq \mathbb{E}_\pi[G_t | S_t = s, A_t = a]$, where $\mathbb{E}_\pi[\cdot]$ denotes the expectation over trajectories induced by $\pi$. Given $q^\pi$, one can define a *greedy* policy $\pi'(s) \in \arg\max_a q^\pi(s, a)$. It is guaranteed that $q^{\pi'}(s, a) \geq q^\pi(s, a), \forall (s, a) \in \mathcal{S} \times \mathcal{A}$. The processes of computing $q^\pi$ and $\pi'$ are known, respectively, as the *policy evaluation* and *policy improvement* steps. Under certain conditions, repeatedly executing the policy evaluating and improvement steps leads to an optimal policy $\pi^*(s) \in \arg\max_a q^*(s, a)$ (Puterman, 2014). Let $(\mathcal{S}, \mathcal{A}, p, \mu, \gamma)$ be a *Markov control process* (MPC) (Puterman, 2014), i.e., an MDP without a reward function. Given an MPC, we define a family of MDPs $\mathcal{M} \triangleq \{(\mathcal{S}, \mathcal{A}, p, r, \mu, \gamma) \mid r : \mathcal{S} \times \mathcal{A} \times \mathcal{S} \to \mathbb{R}\}$ that only differ in their reward function. We refer to any MDP $M \in \mathcal{M}$ (and its corresponding reward function $r$) as a *task*.

### 2.2 Generalized Policy Evaluation and Improvement

Generalized Policy Evaluation (GPE) and Generalized Policy Improvement (GPI) generalize the policy evaluation and improvement steps to the case where an agent has access to a *set* of policies. In particular, GPE and GPI are used, respectively, *(i)* to evaluate a policy on multiple tasks; and *(ii)* to construct a policy, capable of solving a particular novel task, by improving on an existing set of policies.

**Definition 2.1** (Barreto et al. (2020))**.** *GPE is the computation of the action-value function of a policy $\pi$, $q_r^\pi(s, a)$, for a set of tasks. Moreover, given a set of policies $\Pi$ and a reward function of an arbitrary task, $r$, GPI defines a policy, $\pi'$, such that $q_r^{\pi'}(s, a) \geq \max_{\pi \in \Pi} q_r^\pi(s, a)$ for all $(s, a) \in \mathcal{S} \times \mathcal{A}$.*

Based on the latter definition, for any reward function $r$, a *GPI policy* can be constructed based on a set of policies $\Pi$ as

$$\pi^{\text{GPI}}(s; \Pi) \in \arg\max_{a \in \mathcal{A}} \max_{\pi \in \Pi} q_r^\pi(s, a). \tag{1}$$

Let $q_r^{\text{GPI}}(s, a)$ be the action-value function of $\pi^{\text{GPI}}$ under the reward function $r$. The GPI theorem (Barreto et al., 2017) ensures that $\pi^{\text{GPI}}$ in Eq. (1) satisfies Def. 2.1; i.e., that $q_r^{\text{GPI}}(s, a) \geq \max_{\pi \in \Pi} q_r^\pi(s, a)$ for all $(s, a) \in \mathcal{S} \times \mathcal{A}$. This implies that Eq. (1) can be used to define a policy guaranteed to perform at least as well as all other policies $\pi_i \in \Pi$ w.r.t. any given reward function, $r$. The GPI theorem can also be extended to the case $q^{\pi_i}$ is replaced with an approximation, $\tilde{q}^{\pi_i}$ (Barreto et al., 2018).

## 2.3 GPE&GPI via Successor Features

*Successor features* (SFs) allows us to perform GPE&GPI efficiently (Barreto et al., 2017). Assume that the reward functions of interest are linear w.r.t. *reward features*, $\phi : \mathcal{S} \times \mathcal{A} \times \mathcal{S} \to \mathbb{R}^d$. That is, a reward function, $r_{\mathbf{w}}$, can be expressed as $r_{\mathbf{w}}(s, a, s') = \phi(s, a, s') \cdot \mathbf{w}$, where $\mathbf{w} \in \mathbb{R}^d$ is a weight vector. Then, the SFs of a policy $\pi$ for a given state-action pair $(s, a)$, $\boldsymbol{\psi}^\pi(s, a) \in \mathbb{R}^d$, are defined as

$$\boldsymbol{\psi}^\pi(s, a) \triangleq \mathbb{E}_\pi \left[ \sum_{i=0}^{\infty} \gamma^i \boldsymbol{\phi}_{t+i} \mid S_t = s, A_t = a \right], \tag{2}$$

where $\boldsymbol{\phi}_t \triangleq \phi(S_t, A_t, S_{t+1})$. Notice that the definition of SFs corresponds to a form of action-value function, where the features $\boldsymbol{\phi}_t$ play the role of rewards. Thus, SFs can be learned through any temporal-difference (TD) learning algorithm (e.g., Q-learning (Watkins, 1989; Mnih et al., 2015) for learning $\boldsymbol{\psi}^{\pi^*}$). We refer to $\boldsymbol{\psi}^\pi \triangleq \mathbb{E}_{S_0 \sim \mu} [\boldsymbol{\psi}^\pi(S_0, \pi(S_0))]$ as the *SF vector* associated with $\pi$, where the expectation is with respect to the initial state distribution.

Given the SFs $\boldsymbol{\psi}^\pi(s, a)$ of a policy $\pi$, it is possible to *directly* compute the action-value function $q_{\mathbf{w}}^\pi(s, a)$ of $\pi$, under *any* linearly-expressible reward functions, $r_{\mathbf{w}}$, as follows: $q_{\mathbf{w}}^\pi(s, a) = \mathbb{E}_\pi \left[ \sum_{i=0}^{\infty} \gamma^i r_{\mathbf{w}}(S_{t+i}, A_{t+i}, S_{t+i+1}) \mid S_t = s, A_t = a \right] = \boldsymbol{\psi}^\pi(s, a) \cdot \mathbf{w}$. That is, given any set of reward weights, $\{\mathbf{w}_i\}_{i=1}^n$, GPE can be performed efficiently via inner-products between the SFs and the reward weights: $q_{\mathbf{w}_i}^\pi(s, a) = \boldsymbol{\psi}^\pi(s, a) \cdot \mathbf{w}_i$. Note that the value of a policy $\pi$ under any task $\mathbf{w}$ can be expressed as $v_{\mathbf{w}}^\pi = \boldsymbol{\psi}^\pi \cdot \mathbf{w}$. Let $\Pi = \{\pi_i\}_{i=1}^n$ be a set of policies with corresponding SFs $\Psi = \{\boldsymbol{\psi}^{\pi_i}\}_{i=1}^n$. Based on the definition of GPI (Def. 2.1) and the reward decomposition $r_{\mathbf{w}}(s, a, s') = \phi(s, a, s') \cdot \mathbf{w}$, a *generalized policy*, $\pi^{\text{GPI}} : \mathcal{S} \times \mathcal{W} \to \mathcal{A}$, can then be defined as follows:

$$\pi^{\text{GPI}}(s, \mathbf{w}; \Pi) \in \arg\max_{a \in \mathcal{A}} \max_{\pi \in \Pi} \boldsymbol{\psi}^\pi(s, a) \cdot \mathbf{w}. \tag{3}$$

## 2.4 GPI via the Option Keyboard

The *Option Keyboard* (OK) (Barreto et al., 2019, 2020) is a method that generalizes GPI (Eq. (1)) by allowing agents to use a different weight vector at each time step. It does so by learning a meta-policy $\omega : \mathcal{S} \to \tilde{\mathcal{Z}}$ that outputs weights, $\mathbf{z} \in \mathcal{Z}$, [1] used when following the GPI policy $\pi^{\text{GPI}}(s, \mathbf{z}; \Pi)$ at each state $s$. Intuitively, given a set of policies $\Pi$, $\omega$ modulates the behavior induced by $\pi^{\text{GPI}}$ by controlling the preferences $\omega(s)$ over features for each state $s$. The OK policy, $\pi_\omega^{\text{OK}}$, at each state $s$, is given by:

$$\pi_\omega^{\text{OK}}(s; \Pi) \triangleq \arg\max_{a \in \mathcal{A}} \max_{\pi \in \Pi} \boldsymbol{\psi}^\pi(s, a) \cdot \omega(s). \tag{4}$$

Note that, by construction, $\pi_\omega^{\text{OK}}(s; \Pi) = \pi^{\text{GPI}}(s, \omega(s); \Pi)$. Intuitively, in the "keyboard" analogy, $\omega(s)$ (a "chord") combines base policies in $\Pi$ ("keys") to generate more complex behaviors. A key property of OK is that learning a policy over $\mathcal{Z}$ is often easier than over $\mathcal{A}$, as solutions typically involve repeating the same action $\mathbf{z}$ across multiple timesteps, similar to *temporally extended options*.

# 3 Optimal Behavior Basis

In this section, we formalize the problem of identifying a behavior basis—i.e., a set of base policies—that enables the OK to optimally solve all tasks within a given family of MDPs. We are

---

[1] Without loss of generality, let $\mathcal{Z}$ be the space of $d$-dimensional unit vectors, i.e., $\mathcal{Z} = \{\mathbf{z} \in \mathbb{R}^d \mid ||\mathbf{z}||_2 = 1\}$.

interested in incremental methods for constructing a behavior basis, $\Pi_k$, that provably enables the OK to solve any MDP in $\mathcal{M}$. Specifically, we consider approaches where an agent first iteratively learns a behavior basis for solving some subset of tasks, $\mathcal{M}' \subset \mathcal{M}$. Then, the method should be capable of leveraging GPI or OK to identify additional specialized policies for optimally solving novel, unseen tasks $M \notin \mathcal{M}'$. This process should progressively expand the behavior basis until it converges to a small but sufficient set that guarantees zero-shot optimality. The core problem investigated in this paper is, then, how to identify an optimal set of base policies that an agent should learn to facilitate transfer learning within specific classes of MDPs.

Let $\mathcal{M}_{\text{lin}}^{\phi} \subseteq \mathcal{M}$ be the—possibly infinite—set of MDPs associated with all linearly expressible reward functions. This set, typically studied in the SFs literature, can be defined as

$$\mathcal{M}_{\text{lin}}^{\phi} \triangleq \{(\mathcal{S}, \mathcal{A}, p, r_{\mathbf{w}}, \mu, \gamma) \mid r_{\mathbf{w}} = \phi \cdot \mathbf{w}\}. \tag{5}$$

Each element $M \in \mathcal{M}_{\text{lin}}^{\phi}$ (or, equivalently, its weight vector $\mathbf{w}$) is called a *task*. In what follows, we consider weight vectors that induce convex combinations of features; that is, $\mathcal{W} = \{\mathbf{w} \mid \sum_i w_i = 1, w_i \geq 0, \forall i\}$. This is common practice, e.g., in the multi-objective RL literature (Hayes et al., 2022). Our analyses naturally extend to *linear* combinations of features—in App. B, we show how linear-reward MDPs can be transformed into equivalent convex-reward MDPs (w.l.o.g.) so that our results and algorithms apply directly. Existing algorithms capable of optimally solving all tasks in $\mathcal{M}_{\text{lin}}^{\phi}$ often identify a set of policies, $\Pi_k = \{\pi_i\}_{i=1}^{n}$, such that their associated SF vectors, $\Psi = \{\boldsymbol{\psi}^{\pi_i}\}_{i=1}^{n}$, form a *convex coverage set* (CCS) (Alegre et al., 2022). A CCS is a set of SF vectors that allows the optimal value function, $v_{\mathbf{w}}^* = \boldsymbol{\psi}^{\pi_{\mathbf{w}}^*} \cdot \mathbf{w}$, for any given task $\mathbf{w} \in \mathcal{W}$, to be directly identified:[2]

$$\text{CCS} \triangleq \{\boldsymbol{\psi}^{\pi} \mid \exists \mathbf{w} \in \mathcal{W} \text{ s.t. } \forall \pi', \boldsymbol{\psi}^{\pi} \cdot \mathbf{w} \geq \boldsymbol{\psi}^{\pi'} \cdot \mathbf{w}\}. \tag{6}$$

Unfortunately, methods that compute the complete CCS for $\mathcal{M}_{\text{lin}}^{\phi}$ do not scale to complex tasks since its size can grow exponentially with $d$—counting corner weights is equivalent to Vertex Enumeration (Roijers, 2016). Hence, it becomes critical to develop novel techniques capable of recovering all solutions induced by a CCS without incurring the cost of learning the corresponding full set of policies. This could be achieved, for instance, by methods capable of expressing all solutions in a CCS by combining policies in a behavior basis $\Pi_k$ smaller than the CCS; that is, $|\Pi_k| \leq |\text{CCS}|$.

We start by extending the definition of the OK (Eq. (4)) to include the description of the task being solved, $\mathbf{w} \in \mathcal{W}$, as one of its arguments. That is, we consider meta-policies $\omega : \mathcal{S} \times \mathcal{W} \to \mathcal{Z}$, and OK policies defined as

$$\pi_{\omega}^{\text{OK}}(s, \mathbf{w}; \Pi) \triangleq \arg\max_{a \in \mathcal{A}} \max_{\pi \in \Pi} \boldsymbol{\psi}^{\pi}(s, a) \cdot \omega(s, \mathbf{w}). \tag{7}$$

The meta-policy $\omega(s, \mathbf{w})$ enables the OK to decompose the optimal policy for a task $\mathbf{w}$ by dynamically assigning state-dependent linear weights to the SFs of its various base policies. Note that if $\omega(s, \mathbf{w}) = \mathbf{w}$ for all $s \in \mathcal{S}$, we recover GPI. Given an arbitrary set of base policies $\Pi_k$, we define the space of all policies expressible by the OK, $\Pi^{\text{OK}}(\Pi_k)$, and their associated SF vectors, $\Psi^{\text{OK}}(\Pi_k)$, respectively, as

$$\Pi^{\text{OK}}(\Pi_k) \triangleq \{\pi_{\omega}^{\text{OK}}(\cdot, \mathbf{w}; \Pi_k) \mid \mathbf{w} \in \mathcal{W}, \omega : \mathcal{S} \times \mathcal{W} \to \mathcal{Z}\} \text{ and } \Psi^{\text{OK}}(\Pi_k) \triangleq \{\boldsymbol{\psi}^{\pi} \mid \pi \in \Pi^{\text{OK}}(\Pi_k)\}.$$

We are now ready to mathematically define our goal:

**Goal**: Learn a set of policies (the *behavior basis*) $\Pi_k$ and a meta-policy $\omega$ such that **(1)** $|\Pi_k| \leq |\text{CCS}|$; and **(2)** $\pi_{\omega}^{\text{OK}}(\cdot; \Pi_k)$ is optimal for any task $M \in \mathcal{M}_{\text{lin}}^{\phi}$. The latter condition implies that $\text{CCS} \subseteq \Psi^{\text{OK}}(\Pi_k)$; i.e., the OK is at least as expressive as a CCS.

As observed by Barreto et al. (2019, 2020), learning a meta-policy, $\omega$, is often easier than learning base policies over the original action space, $\mathcal{A}$. Consider, e.g., that if a task $\mathbf{w} \in \mathcal{W}$ can be solved by switching between two base policies in $\Pi_k$, then an optimal meta-policy for $\mathbf{w}, \omega(\cdot, \mathbf{w})$, only needs to output two vectors $\mathbf{z}$ (one for each base policy) for any given state. Thus, solving the goal above is bound to be more efficient than constructing a complete CCS since it requires learning fewer optimal base policies.

---

[2]A CCS may not be unique since tasks can have multiple optimal policies with distinct SF vectors. In what follows, CCS refers specifically to the minimal set satisfying Eq. (6), which is uniquely defined (Roijers et al., 2013).

# 4 Constructing an Optimal Behavior Basis

We now introduce **O**ption **K**eyboard **B**asis (**OKB**), a novel method to solve the goal introduced in the previous section. The OKB (Alg. 1) learns a set of base policies, $\Pi_k$, and a meta-policy, $\omega$, such that the induced OK policy $\pi_\omega^{\text{OK}}(\cdot; \Pi_k)$ is provably optimal w.r.t. any given task $\mathbf{w} \in \mathcal{W}$.

The algorithm starts with a single base policy optimized to solve an arbitrary initial task (e.g., $\mathbf{w} = [1/d, ..., 1/d]^\top$) in its set of policies $\Pi$ (lines 1–2). OKB's initial partial CCS, $\Psi^{\text{OK}}$, and weight support set, $\mathcal{W}^{\text{sup}}$, are initialized as empty sets (line 3). $\mathcal{W}^{\text{sup}}$ will store the weights of tasks the meta-policy has been trained on. At each iteration $k$, OKB carefully selects the weight vectors on which its meta-policy, $\omega$, will be trained so that the policies expressible by $\Psi^{\text{OK}}$ iteratively approximate a CCS. This process is implemented by OK-LS (Alg. 2), an algorithm inspired by the SFs Optimistic Linear Support (SFOLS) method (Alegre et al., 2022) but that operates over a meta-policy $\omega$ rather than the space of base policies.

---

**Algorithm 1: O**ption **K**eyboard **B**asis (OKB)

---

**Input:** MPC with features $\phi(s, a) \in \mathbb{R}^d$

1   $\pi_\mathbf{w}, \psi^{\pi_\mathbf{w}} \leftarrow \texttt{NewPolicy}(\mathbf{w} = \texttt{InitialTask}())$
2   $\Pi_0 \leftarrow \{\pi_\mathbf{w}\}; \Psi_0^{\text{base}} \leftarrow \{\psi^{\pi_\mathbf{w}}\}$
3   $\Psi_0^{\text{OK}} \leftarrow \{\}; \mathcal{W}_0^{\text{sup}} \leftarrow \{\}$
4   Initialize meta-policy $\omega_0$
5   **for** $k = 0, 1, 2, \ldots$ **do**
     ▷ Update meta-policy $\omega$
6      $\omega_{k+1}, \Psi_{k+1}^{\text{OK}}, \mathcal{W}_{k+1}^{\text{sup}} \leftarrow$
      $\texttt{OK-LS}(\omega_k, \Psi_k^{\text{OK}}, \mathcal{W}_k^{\text{sup}}, \Pi_k)$
7      $\mathcal{C} \leftarrow \{\}$
8      **for** $\mathbf{w} \in \texttt{CornerW}(\Psi_k^{\text{base}}) \cup \texttt{CornerW}(\Psi_{k+1}^{\text{OK}})$ **do**
       ▷ Check if task $\mathbf{w}$ is solvable with OK and $\Pi_k$
9        **if** $\pi_\mathbf{w}^*$ *is not in* $\Pi^{\text{OK}}(\Pi_k)$ **then**
10         $\mathcal{C} \leftarrow \mathcal{C} \cup \{\mathbf{w}\}$
11      **if** $\mathcal{C}$ *is empty* **then**
       ▷ Found optimal basis $\Pi$ and meta-policy $\omega$
12        **return** $\omega_{k+1}, \Pi_k, \Psi_{k+1}^{\text{OK}}$
13      **else**
       ▷ Learn a new base policy
14        $\mathbf{w} \leftarrow$ select a task from $\mathcal{C}$
15        $\pi_\mathbf{w}, \psi^{\pi_\mathbf{w}} \leftarrow \texttt{NewPolicy}(\mathbf{w})$
16        $\Pi_{k+1} \leftarrow \Pi_k \cup \{\pi_\mathbf{w}\}$
17        $\Psi_{k+1}^{\text{base}} \leftarrow \Psi_k^{\text{base}} \cup \{\psi^{\pi_\mathbf{w}}\}$
18        $\Pi_{k+1}, \Psi_{k+1}^{\text{base}} \leftarrow$
        $\texttt{RemoveDominated}(\Pi_{k+1}, \Psi_{k+1}^{\text{base}})$

---

**Algorithm 2: OK - Linear Support (OK-LS)**

---

**Input:** Meta-policy $\omega$, partial CCS $\Psi^{\text{OK}}$, weight support $\mathcal{W}^{\text{sup}}$, base policies $\Pi_k$.

1   **while** True **do**
2      $\mathcal{W}^{\text{corner}} \leftarrow \texttt{CornerW}(\Psi^{\text{OK}}) \setminus \mathcal{W}^{\text{sup}}$
3      **if** $\mathcal{W}^{cw}$ *is empty* **then**
4        **return** $\omega, \Psi^{OK}, \mathcal{W}^{sup}$
5      $\mathcal{W}^{\text{sup}} \leftarrow \mathcal{W}^{\text{sup}} \cup \mathcal{W}^{\text{corner}}$
6      $\omega \leftarrow \texttt{TrainOK}(\omega, \mathcal{W}^{\text{sup}}, \Pi_k)$    ▷ (Alg. 3)
7      $\Psi^{\text{OK}} \leftarrow \{\psi^{\pi_\omega^{\text{OK}}(\cdot, \mathbf{w}; \Pi_k)} \mid \mathbf{w} \in \mathcal{W}^{\text{sup}}\}$
8      $\Psi^{\text{OK}} \leftarrow \texttt{RemoveDominated}(\Psi^{\text{OK}})$

---

Next, in lines 8–10, OKB identifies a set of candidate tasks, $\mathcal{C}$, that the OK cannot optimally solve with its current set of policies, $\Pi_k$; i.e., a set of tasks for which $\pi_\mathbf{w}^*$ is not included in $\Pi^{\text{OK}}(\Pi_k)$. We discuss how to check this condition in Section 4.1 If $\mathcal{C}$ is empty (line 11), the OKB terminates and returns a meta-policy ($\omega$) and base policies ($\Pi_k$) capable of ensuring that CCS $\subseteq \Psi^{\text{OK}}(\Pi_k)$. This is due to Thm. 4.3, which we discuss later. If $\mathcal{C}$ is not empty, OKB selects a task from it and adds a new corresponding optimal base policy to its behavior basis, $\Pi_k$ (lines 14–17). In line 18, RemoveDominated removes redundant policies—policies that are not strictly required to solve at least one task.

In Algorithms 1 and 2, the function $\texttt{CornerW}(\Psi)$ takes a set of SF vectors, $\Psi$, as input and returns a set of *corner weights* (see App. F.1). Intuitively, both Alg. 1 and Alg. 2 rely on the fact that *(i)* a task $\mathbf{w} \in \mathcal{W}$ that maximizes $\Delta(\mathbf{w}) \triangleq v_\mathbf{w}^* - \max_{\pi \in \Pi_k} v_\mathbf{w}^\pi$ is guaranteed to be a corner weight, and *(ii)* if OKB has an optimal policy for all corner weights, then it must have identified a CCS. These results were shown in prior work on constructing a CCS (Roijers, 2016; Alegre et al., 2022) and can also be proven using the fundamental theorem of linear programming.

OK-LS (Alg. 2) trains the meta-policy, $\omega$, on selected tasks $\mathcal{W}^{\text{cw}}$ (line 2) using the base policies $\Pi_k$, so that is partial CCS, $\Psi^{\text{OK}}$, iteratively approximates a CCS. It stores the tasks $\mathbf{w}$ it has already trained on, as well as the corner weights of the current iteration, in the weight support set, $\mathcal{W}^{\text{sup}}$ (line 5). If $\mathcal{W}^{\text{cw}}$ is empty in a given iteration, no tasks remain to be solved, and the algorithm returns the updated meta-policy. Otherwise, OK-LS adds the corner weights $\mathcal{W}^{\text{cw}}$ to $\mathcal{W}^{\text{sup}}$ and trains the meta-policy $\omega$ on the tasks $\mathbf{w} \in \mathcal{W}^{\text{sup}}$ using the TrainOK subroutine (Alg. 3). Finally, in line 7, OK-LS computes the SF vectors of the policies induced by OK for each $\mathbf{w} \in \mathcal{W}^{\text{sup}}$, updating its partial CCS, $\Psi^{\text{OK}}$.[3] In App. D, we discuss how to train $\omega$

---

[3] Notice that SFs only need to be computed for vectors newly added to $\mathcal{W}^{\text{sup}}$ in the current iteration (lines 6-7).

using an actor-critic RL method (Alg. 3). Finally, note that to accelerate the learning of $\omega$, the corner weights in $\mathcal{W}^{\text{cw}}$ can be prioritized, similar to Alegre et al. (2022).

## 4.1 Condition for OK Optimality

In this section, we address the following question: *Given an arbitrary task with reward function $r$ and a set of base policies $\Pi_k$, is $\pi_r^* \in \Pi^{\text{OK}}(\Pi_k)$?* In other words, does a meta-policy $\omega$ exist such that the OK policy, $\pi^{\text{OK}}\omega(\cdot; \Pi_k)$, can represent the optimal policy $\pi_r^*$? Determining whether this holds is essential for implementing the OKB step in line 9.

**Proposition 4.1.** *Let $\Pi_k = \{\pi_i\}_{i=1}^n$ be a set of base policies with corresponding SFs $\Psi = \{\boldsymbol{\psi}^{\pi_i}\}_{i=1}^n$. Given an arbitrary reward function $r$, an optimal OK policy $\pi_\omega^{OK}(\cdot; \Pi)$ can only exist if there exists an OK meta-policy, $\omega : \mathcal{S} \to \mathcal{Z}$, such that for all $s \in \mathcal{S}$,*

$$\arg\max_{a \in \mathcal{A}} \max_{\pi \in \Pi_k} \boldsymbol{\psi}^\pi(s, a) \cdot \omega(s) = \arg\max_{a \in \mathcal{A}} q_r^*(s, a). \tag{8}$$

This proposition provides a sufficient condition for the existence of an optimal OK meta-policy $\omega$ for a given reward function $r$. Intuitively, it implies that the OK policy should be able to express all optimal actions through $\omega$ and $\Psi$. Next, we show how to verify this condition without requiring access to $q_r^*$.

Let $\Pi_k$ be a set of base policies, $\omega$ be an OK meta-policy, and $q_r^\omega(s, \mathbf{z}) = r(S_t, A_t) + \gamma \mathbb{E}_\omega[q_r^\omega(S_{t+1}, \omega(S_{t+1})) \mid S_t = s, A_t = \pi^{\text{GPI}}(S_t, \mathbf{z}; \Pi_k)]$ be the meta-policy's action-value function for task $r$. Given a state-action pair $(s, a)$, let the *advantage function* of $\omega$ for executing action $a$ in state $s$ be

$$A_r^\omega(s, a) \triangleq r(s, a) + \gamma \mathbb{E}_\omega[q_r^\omega(S_{t+1}, \omega(S_{t+1})) \mid S_t{=}s, A_t{=}a] - q_r^\omega(s, \omega(s)). \tag{9}$$

It is well-known that an optimal policy's advantage function is zero when evaluated at an optimal action; i.e., $A^{\pi^*}(s, a^*) = 0$ for all $s \in \mathcal{S}$. Thm. 4.2 uses this insight to introduce a principled way to verify if $\pi_r^* \in \Pi^{\text{OK}}(\Pi_k)$.

**Theorem 4.2.** *Let $\Pi_k$ be a set of base policies and $\omega^*$ be a meta-policy trained to convergence to solve a given task $r$. If $A_r^{\omega^*}(s, a) > 0$ for some $(s, a)$, then action $a$ is not expressible by $\pi_{\omega^*}^{OK}(\cdot; \Pi_k)$. Consequently, the OK with base policies $\Pi_k$ cannot represent an optimal policy for $r$.*

Note that the procedure for verifying whether the OK is optimal for a given task $r$ (Thm. 4.2) is a sufficient, but not necessary, condition to guarantee optimality. It serves two key purposes: *(i)* to avoid training on tasks $\omega$ whose optimal policies have either already been learned or that can be reconstructed from $\Pi_k$ and $\omega$, and *(ii)* to prioritize which tasks to train on in order to accelerate learning. In other words, the procedure for assessing OK optimality is primarily a tool to reduce training effort; our method does not rely on an exact test to ensure the correctness of the behavior basis it constructs. For a detailed discussion of how Thm. 4.2 can be applied in practical implementations of OKB to verify whether $\pi_r^* \in \Pi^{\text{OK}}(\Pi_k)$, see App. E.

## 4.2 Theoretical Results

In this section, we present the theoretical guarantees of OKB (Alg. 1). Proofs of the theorems can be found in App. A.

**Theorem 4.3.** *OKB returns a set of base policies, $\Pi_k$, and a meta-policy, $\omega$, such that $|\Pi_k| \leq |\text{CCS}|$ and $\pi_\omega^{OK}(\cdot; \Pi_k)$ is optimal for any task $M \in \mathcal{M}_{lin}^\phi$. This implies that $\text{CCS} \subseteq \Psi^{OK}(\Pi_k)$; i.e., the OK matches or exceeds the expressiveness of a CCS and ensures zero-shot optimality for all $M \in \mathcal{M}_{lin}^\phi$.*

This is a key result of this paper: OKB can provably identify a behavior basis, $\Pi_k$, that enables an option keyboard to *optimally solve* any task $M \in \mathcal{M}_{\text{lin}}^\phi$. Furthermore, it is capable of ensuring zero-shot optimality using a set of base policies potentially smaller than a CCS. This is the first method to offer such a guarantee—until now, all existing approaches required the number of base policies to be *strictly equal* to the size of the CCS. This limitation posed a major scalability challenge: computing a full CCS becomes intractable as the number of reward features $d$ grows, since its size can grow exponentially with $d$—counting corner weights is equivalent to Vertex Enumeration (Roijers, 2016). OKB overcomes this bottleneck by avoiding the need to construct a full CCS, thereby reducing the total cost of computing optimal policies for all $M \in \mathcal{M}_{\text{lin}}^\phi$. See below for more details.

**Proposition 4.4.** *Let $\Pi_k$ be the set of policies learned by OKB (Alg. 1) when solving tasks that are linearly expressible in terms of reward features $\phi(s, a) \in \mathbb{R}^d$. Let $r$ be an arbitrary reward function that is non-linear with respect to $\phi$. Suppose the optimal policy for $r$ can be expressed by alternating, as a function of the state, between policies in the CCS induced by $\phi$. That is, suppose that for all $s \in \mathcal{S}$, there exists a policy $\pi_{\mathbf{w}}^*$ (optimal for some $\mathbf{w} \in \mathcal{W}$) such that $\pi_r^*(s) = \pi_{\mathbf{w}}^*(s)$. Then, the OK can represent the optimal policy for $r$ using the set of base policies $\Pi_k$; i.e., $\pi_r^* \in \Pi^{\text{OK}}(\Pi_k)$.*

This deepens the theoretical understanding of the OKB by showing it is strictly more expressive than existing methods. In particular, it allows us to characterize **(1)** conditions under which the OKB matches the expressiveness of a CCS while relying on strictly *fewer* policies; and **(2)** a broad class of *non-linear* tasks that can be optimally solved by an OK using the behavior basis learned by OKB.

First, Prop. 4.4 establishes the optimality of the OKB in transfer learning scenarios where related tasks can be solved by reusing parts of existing solutions—that is, the widely studied setting in which transfer is possible due to shared structure across tasks. Concretely, the OKB is guaranteed to optimally solve, in a zero-shot manner, tasks whose solutions decompose into sequences of sub-policies. *Building on this result and Thm. 4.3, we formally characterize, in App. A.5, the conditions under which the OKB is as expressive as a CCS while relying on strictly **fewer** policies.*

Second, Proposition 4.4 precisely characterizes a class of non-linear tasks that can be optimally solved by an OK using the behavior basis learned by OKB. Intuitively, OKB can solve any task whose optimal policy can be constructed by alternating between the optimal policies for tasks in $\mathcal{M}_{\text{lin}}^{\phi}$, even when the task itself is non-linear. *This implies, importantly, that while existing methods that compute a CCS are limited to linearly expressible tasks, OKB can handle a broader class of non-linear tasks.* Furthermore, as discussed above, the OKB achieves this *without* having to learn all policies in the CCS. We further discuss the properties of OK under non-linear reward functions in App. C.

To state our next theoretical result, we first recall Thm. 2 by Barreto et al. (2017):

**Theorem 4.5** (Barreto et al. (2017))**.** *Let $\Pi = \{\pi_i\}_{i=1}^n$ be a set of optimal policies w.r.t. tasks $\{\mathbf{w}_i\}_{i=1}^n$. Let $\{\hat{\psi}^{\pi_i}\}_{i=1}^n$ be approximations to their SFs and $\mathbf{w} \in \mathcal{W}$ be a task. Let $|q_{\mathbf{w}}^{\pi_i}(s, a) - \hat{q}_{\mathbf{w}}^{\pi_i}(s, a)| \leq \epsilon$ for all $(s, a) \in \mathcal{S} \times \mathcal{A}$, and $\pi_i \in \Pi$. Let $\phi_{\max} \triangleq \max_{s,a} ||\phi(s, a)||$. Then, it holds that:*

$$q_{\mathbf{w}}^*(s, a) - q_{\mathbf{w}}^{\pi^{GPI}}(s, a) \leq \frac{2}{1-\gamma}\left(\phi_{\max}\min_i ||\mathbf{w} - \mathbf{w}_i|| + \epsilon\right) \quad \textit{for all } (s, a) \in \mathcal{S} \times \mathcal{A}. \tag{10}$$

Thm 4.5 describes the optimality gap of GPI policies, how it depends on the available base policies, and how it is affected by function approximation errors. It does not, however, characterize the optimality gap of option keyboards, which generalize GPI policies. We introduce and highlight two generalizations of this theorem: **(1)** Since the OK generalizes GPI policies, it follows that $q_{\mathbf{w}}^*(s, a) - q_{\mathbf{w}}^{\pi_{\omega}^{\text{OK}}}(s, a) \leq q_{\mathbf{w}}^*(s, a) - q_{\mathbf{w}}^{\pi^{\text{GPI}}}(s, a)$; and **(2)** when Eq. (8) (Prop. 4.1) is satisfied, $q_{\mathbf{w}}^*(s, a) - q_{\mathbf{w}}^{\pi_{\omega}^{\text{OK}}}(s, a) = 0$. That is, OK can entirely avoid optimality gaps from approximation errors in the base policies' SFs. Thus, OK and OKB are not only more expressive than GPI (Thm. 4.3) but also naturally lead to transfer learning strategies that are significantly more robust to approximation errors than GPI.

## 5 Experiments

We now empirically evaluate OKB and investigate the following research questions: **Q1**: Can OKB approximate a CCS more effectively while requiring fewer base policies than competing methods? **Q2**: Does OKB's performance advantage grow with problem complexity, as measured by the number of reward features $d$? **Q3**: Can the base policies learned by OKB be used to solve tasks with non-linear reward functions, under the conditions in Prop. 4.4?

Fig. 1 depicts the domains used in our experiments. To handle the high-dimensional state space of these domains, we learn base policies using a Universal SF

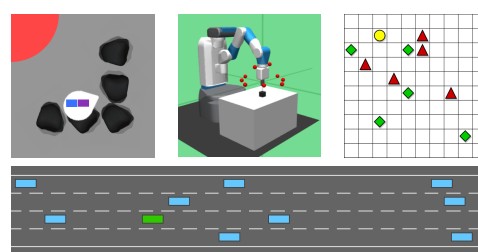

Figure 1: Domains used in the experiments: Minecart, FetchPickAndPlace, Item Collection, and Highway.

Approximator (USFA) (Borsa et al., 2019). Furthermore, rather than learning a separate SF for each base policy $\pi \in \Pi_k$, we train a single USFA, $\boldsymbol{\psi}(s, a, \mathbf{w})$, conditioned on task vectors $\mathbf{w}$, such that $\pi_{\mathbf{w}}(s) \approx \arg\max_{a \in \mathcal{A}} \boldsymbol{\psi}(s, a, \mathbf{w}) \cdot \mathbf{w}$. Each call to NewPolicy($\mathbf{w}$) in Alg.1 trains the USFA to solve task $\mathbf{w}$. We employ USFAs in our experiments primarily to avoid the memory and computational costs of training a separate neural network for each policy's SFs. Moreover, USFAs typically achieve a level of approximation comparable to—or better than—that of independently trained networks, since in both cases generalization depends mainly on the architecture, capacity, and distribution of training data. See App. F for more details.

In all experiments, we report the mean normalized return of each method (normalized with respect to the minimum and maximum returns observed for a given task) along with the 95% bootstrapped confidence interval over 15 random seeds. We compare OKB to SFOLS (Alegre et al., 2022), a method that identifies a CCS using GPI policies, and to OKB-Uniform, a variant of OKB that selects tasks uniformly from $\mathcal{W}$ in line 14 of Alg. 1, rather than from $\mathcal{C}$. An iteration corresponds to each method training a base policy within a fixed budget of environment interactions. To ensure fair comparisons, since OKB must also train a meta-policy while SFOLS does not, we restrict OKB to using only half of its budget for learning base policies, allocating the other half to training the meta-policy (Alg. 2). This makes the comparison more conservative for OKB, as it has fewer interactions available to learn base policies.

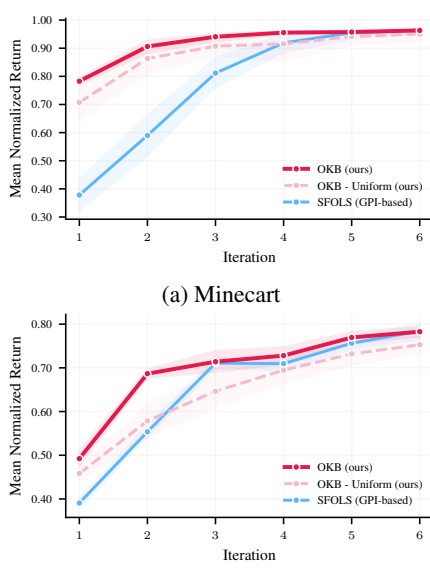

(a) Minecart

(b) Highway

Figure 2: Mean normalized return per iteration for each method on a set of test tasks.

Fig. 2 depicts each method's mean return over a set of test tasks, $M \in \mathcal{M}_{\text{lin}}^{\phi}$,[4] as a function of the iteration number (i.e., the number of base policies learned). We report results for the Minecart domain—a classic multi-objective RL problem—and the Highway domain. Both domains have reward functions defined by $d = 3$ reward features, which are detailed in App. F.2. These results demonstrate that OKB achieves strong performance across test tasks with a small number of base policies, positively answering research question Q1: OKB can approximate a CCS more effectively using fewer base policies. This is particularly evident in Fig. 2a, where OKB reaches near-optimal performance with just 2–3 base policies. While OKB-Uniform also performs well in the Minecart domain, its lower performance in the Highway domain (Fig. 2b) highlights the importance of expanding the set of base policies by carefully selecting promising tasks—defined by corner weights—for training. Across both domains, OKB consistently outperforms all competing methods.

To investigate **Q2**, we evaluate each method in the FetchPickAndPlace domain with varying numbers of reward features, $d \in \{2, 4, 6, 8\}$. In this domain, an agent controls a robotic arm that must grasp a block and move it to a specified location. Each of the $d$ reward features represents the block's distance to a different target location (red spheres in Fig. 1). Fig. 3 shows that as the number of target locations ($d$) increases, the performance gap between OKB and SFOLS increases significantly—positively answering **Q2**.[5] Intuitively, OKB focuses on tasks where the OK cannot express optimal actions (see Thm. 4.2). By learning the corresponding optimal policies, OKB quickly identifies a behavior basis that enables the OK to solve tasks across the entire space of task vectors $\mathcal{W}$. Conversely, the gap between OKB and OKB-Uniform decreases since a larger number of reward features enhances the expressivity of the OK (Prop. 4.1). The total computation time required by OKB to train a meta-policy and base policies for zero-shot optimality

---

[4]To generate test task sets $\mathcal{W}' \subset \mathcal{W}$ for different values of $d$, we used the method introduced by Takagi et al. (2020), which produces uniformly spaced weight vectors in $\mathcal{W}$.

[5]Importantly, to the best of our knowledge, most state-of-the-art methods are typically evaluated on problems with dimensionality at most $d = 4$ (e.g., SFOLS and SIP). In our experiments, by contrast, we double the number of reward features, thus showing that our method remains effective and continues to outperform all baselines even on problems an order of magnitude more challenging.

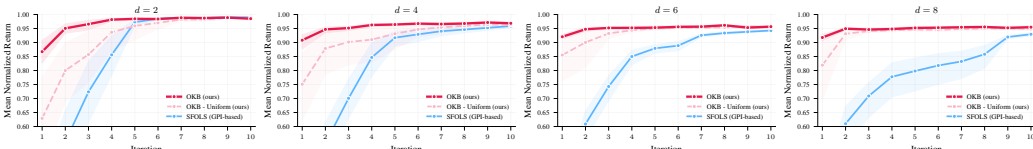

Figure 3: Mean normalized return over test tasks as a function of iteration number (i.e., number of base policies learned per method) [FetchPickAndPlace]. As $d$ increases, OKB's performance advantage over SFOLS (a state-of-the-art GPI-based algorithm) grows.

is consistently comparable to or lower than that of competing methods, as it learns and evaluates fewer policies at inference time.

Next, we investigate **Q3** by conducting an experiment similar to the one proposed by Alver and Precup (2022). We compare OKB to relevant competitors in an environment with non-linear reward functions. The Item Collection domain (Fig. 1; top right) is a $10 \times 10$ grid world where agents must collect two types of items. Reward features are indicator functions that signal whether the agent has collected a particular item in the current state. This domain requires function approximation due to the combinatorial number of possible states. After OKB learns a behavior basis $\Pi_k$, the agent trains a meta-policy $\omega : \mathcal{S} \to \mathcal{Z}$ to solve a task with a *non-linear reward function*. Specifically, this is a sequential task where the agent must collect all instances of one item type before collecting any items of another type. We compare OKB with SIP (Alver and Precup, 2022),[6] a method that learns $d$ base policies—each maximizing a specific reward feature—and a meta-policy $\omega$ over a discrete set of weights in $\mathcal{W}$. We also compare OKB against a baseline DQN agent that learns tasks from scratch and against GPI policies (horizontal blue lines), which are optimal for either exclusively prioritizing one item type or assigning equal importance to both. Fig. 4 shows the mean total reward achieved by each method in this non-linear task as a function of the total number of environment interactions required to train the meta-policies. Both OKB and SIP successfully solved the task, with OKB doing so more quickly. The DQN baseline failed to solve the task as directly learning a policy over $\mathcal{A}$ is significantly more difficult.

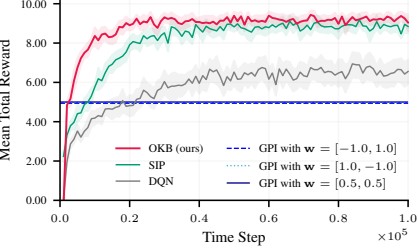

Figure 4: Mean return in the Item Collection domain under a *non-linear* reward function.

Figure 5: Continuous actions from the meta-policy on a sample task. OKB selects base policies in a temporally consistent way.

Finally, we examine the qualitative behavior of OKB policies by visualizing the continuous actions produced by the meta-policy $\omega(s, \mathbf{w})$ while solving a sample task in the Highway domain, where the agent must prioritize driving fast and staying in the rightmost lane. In Fig. 5, the color of each timestep's column represents the selected base policy. The agent initially accelerates as quickly as possible by following base policy $\pi_1$ for the first 10 timesteps while the rightmost lane is occupied. At timestep $t = 10$, the agent turns right and switches to base policy $\pi_2$, which is optimized for driving in the rightmost lane. At $t = 20$, $\pi_1$ becomes active again, allowing the agent to accelerate while staying in the rightmost lane. Note that while the meta-policy $\omega$ is relatively smooth, the base policies it selects are complex. In particular, although $\mathbf{z}_t$ remains approximately constant during the first 10 timesteps, the underlying primitive actions $a \in \mathcal{A}$ (Fig. 5; black numbers at the top of each column) continuously alternate between turning left or right, accelerating, and idling. Finally, notice that OKB selects base policies in a temporally consistent manner, suggesting that it learns to identify temporally extended behaviors—akin to *options* (Sutton et al., 1999)—that help solve the task.

---

[6]SIP was not part of previous experiments as it assumes independent reward features (Alver and Precup, 2022).

# 6   Related Work

**OK and GPI.** Previous works have extended the OK in different ways. Carvalho et al. (2023b) introduced a neural network architecture for jointly learning reward features and SFs, while Machado et al. (2023) proposed using the OK with base policies that maximize Laplacian-based reward features. However, unlike our work, these methods do not provide theoretical guarantees on optimality for solving specific families of tasks. Other works have introduced GPI variants for risk-aware transfer (Gimelfarb et al., 2021), mitigating function approximation errors (Kim et al., 2022), planning with approximate models (Alegre et al., 2023a), planning to solve tasks defined by finite state automata (Kuric et al., 2024), and combining non-Markovian policies (Thakoor et al., 2022). These approaches improve GPI in ways that are orthogonal to our contribution and could potentially be combined with our method.

**Learning behavior basis.** Previous works have addressed the problem of learning an effective behavior basis for transfer learning within the SF framework (Nemecek and Parr, 2021; Zahavy et al., 2021). The method introduced by Alver and Precup (2022) assumes that reward features are independent (i.e., can be controlled independently) and that the MDP's transition function is deterministic. These assumptions often do not hold in complex RL domains, including most of those studied in Section 5. Alegre et al. (2022) proposed a method that learns a set of policies corresponding to a CCS and combines them with GPI. In contrast, we use option keyboards, which enable a broader range of policies to be expressed, allowing our method to approximate a CCS more effectively with a smaller behavior basis.

**Learning features.** While we focused on identifying an optimal behavior basis given a particular set of reward features, other works have addressed the complementary problem of learning more expressive reward features (Touati and Ollivier, 2021; Carvalho et al., 2023a; Chua et al., 2024). A promising future research direction is to combine OKB with methods such as the Forward-Backward representation (Touati et al., 2023) to construct an optimal behavior basis under learned reward features.

# 7   Conclusions

We introduced OKB, a principled method with strong theoretical guarantees for identifying the optimal behavior basis for the Option Keyboard (OK). Our theoretical results, supported by thorough empirical analysis, show that OKB significantly reduces the number of base policies required to achieve zero-shot optimality in new tasks compared to state-of-the-art methods. In particular, OKB constructs a behavior basis efficiently by carefully selecting base policies that iteratively improve its meta-policy's approximation of the CCS. We empirically evaluate OKB in challenging high-dimensional RL problems, demonstrating that it consistently outperforms GPI-based approaches. Notably, its performance advantage becomes increasingly pronounced as the number of reward features grows. Finally, we prove that OKB's expressivity surpasses that of a CCS, enabling it to optimally solve specific classes of non-linear tasks. Among several possible research directions, one promising direction is to extend OKB with temporally extended meta-policies that incorporate learned termination conditions. The resulting expressiveness could further expand OKB's flexibility and efficiency in reusing behaviors across tasks. Furthermore, recently identified mathematical connections between multi-objective RL (MORL) and SFs (Alegre et al., 2022) open the door to adapting OKB to integrate with MORL techniques, paving the way for unified representations that improve both sample efficiency and generalization across objectives.

## Acknowledgments

This work is partially supported by CAPES (Coordenação de Aperfeiçoamento de Pessoal de Nível Superior - Brazil, Finance Code 001). Ana Bazzan is partially supported by CNPq under grant number 304932/2021-3.

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

# Appendix

## A Proofs

### A.1 Proof of Proposition 4.1

**Proposition** (4.1). *Let $\Pi_k = \{\pi_i\}_{i=1}^n$ be a set of base policies with corresponding SFs $\Psi = \{\psi^{\pi_i}\}_{i=1}^n$. Given an arbitrary reward function $r$, an optimal OK policy $\pi_\omega^{OK}(\cdot; \Pi)$ can only exist if there exists an OK meta-policy, $\omega : \mathcal{S} \to \mathcal{Z}$, such that for all $s \in \mathcal{S}$,*

$$\arg\max_{a \in \mathcal{A}} \max_{\pi \in \Pi_k} \psi^\pi(s, a) \cdot \omega(s) = \arg\max_{a \in \mathcal{A}} q_r^*(s, a). \tag{11}$$

*Proof.* The proof follows directly from the definition of a deterministic optimal policy for a given reward function $r$. Recall that $\pi_r^*(s) \in \arg\max_{a \in \mathcal{A}} q_r^*(s, a)$ and $\pi_\omega^{OK}(s; \Pi) \in \arg\max_{a \in \mathcal{A}} \max_{\pi \in \Pi} \psi^\pi(s, a) \cdot \omega(s)$. If for some state $s \in \mathcal{S}$, there is no value of $\mathbf{z} = \omega(s)$ such that Eq. (11) holds, then it is not possible to define a meta policy $\omega$ such that $\pi_\omega^{OK}(s; \Pi) = \pi_r^*(s)$. $\square$

### A.2 Proof of Theorem. 4.2

**Theorem** (4.2). *Let $\Pi_k$ be a set of base policies and $\omega^*$ be a meta-policy trained to convergence to solve a given task $r$. If $A_r^{\omega^*}(s, a) > 0$ for some $(s, a)$, then action $a$ is not expressible by $\pi_{\omega^*}^{OK}(\cdot; \Pi_k)$. Consequently, the OK with base policies $\Pi_k$ cannot represent an optimal policy for $r$.*

*Proof.* First, recall the definitions of $q_r^\omega(s, \mathbf{z})$ and $A_r^\omega(s, a)$:

$$q_r^\omega(s, \mathbf{z}) \triangleq r(S_t, A_t) + \gamma \mathbb{E}_\omega[q_r^\omega(S_{t+1}, \omega(S_{t+1})) \mid S_t = s, A_t = \pi^{GPI}(S_t, \mathbf{z}; \Pi_k)], \tag{12}$$

$$A_r^\omega(s, a) \triangleq r(S_t, A_t) + \gamma \mathbb{E}_\omega[q_r^\omega(S_{t+1}, \omega(S_{t+1})) \mid S_t = s, A_t = a] - q_r^\omega(s, \omega(s)). \tag{13}$$

$A_r^\omega(s, a)$ measures the relative benefit of executing action $a$ in state $s$ compared to following the current OK policy $\pi_\omega^{OK}$.

If $A_r^\omega(s, a) > 0$, then executing $a$ in $s$ provides a *higher expected return* than the action currently selected by $\pi_\omega^{OK}$. Hence, if there exists an $(s, a)$ such that $A_r^\omega(s, a) > 0$, this implies that the OK policy fails to represent at least one optimal action, meaning it is suboptimal.

Recall from Prop. 4.1 that for the OK policy to be optimal, it must express the optimal action $a^*$ for every $s$:

$$\arg\max_{a \in A} \max_{\pi \in \Pi_k} \psi^\pi(s, a) \cdot \omega(s) = \arg\max_{a \in A} q_r^*(s, a).$$

Since $\omega^*$ is trained until convergence to solve $r$, it has reached a stable policy. However, if there exists an $(s, a)$ such that $A_r^{\omega^*}(s, a) > 0$, then

$$q_r^*(s, a) > q_r^{\omega^*}(s, \omega^*(s)).$$

This means that in at least one state-action pair, the OK policy does not select an action that maximizes the expected return. Since $\pi_{\omega^*}^{OK}$ is restricted to the actions expressible via its base policies $\Pi_k$, this implies that

$$a^* \notin \left\{ \arg\max_a \max_{\pi \in \Pi_k} \psi^\pi(s, a) \cdot \omega^*(s) \right\}.$$

Thus, **no possible weighting of the base policies can recover the optimal action** in state $s$.

Since at least one optimal action $a^*$ is missing from the action space of the OK policy, it follows that $\pi_{\omega^K}^{OK}$ is not an optimal policy for $r$, and the task $r$ cannot be optimally solved using the given base policies $\Pi_k$. Therefore, to guarantee optimality, additional base policies must be introduced into $\Pi_k$. $\square$

### A.3 Proof of Theorem. 4.3

**Lemma A.1** (Alegre et al. (2022)). *Let $\Psi_k \subseteq \mathrm{CCS}$ be a subset of the CCS. If, for all corner weights $\mathbf{w} \in \mathrm{CornerW}(\Psi_k)$, it holds that $\max_{\boldsymbol{\psi}^\pi \in \Psi_k} \boldsymbol{\psi}^\pi \cdot \mathbf{w} = v_{\mathbf{w}}^*$, then $\Psi_k$ contains all elements of the CCS, i.e., $\mathrm{CCS} \subseteq \Psi_k$.*

This lemma follows directly from the theoretical guarantees of previous methods that construct a CCS, e.g., OLS and SFOLS (Roijers, 2016; Alegre et al., 2022). Intuitively, this result states that to check if a given set of SF vectors, $\Psi_k$, is a CCS, it is sufficient to check if it contains an optimal policy for all of its corner weights. If this is the case, then it is not possible to identify some alternative weight vector $\mathbf{w} \in \mathcal{W}$ for which it does not contain an optimal policy.

**Theorem** (4.3). *OKB returns a set of base policies, $\Pi_k$, and a meta-policy, $\omega$, such that $|\Pi_k| \leq |\mathrm{CCS}|$ and $\pi_\omega^{OK}(\cdot; \Pi_k)$ is optimal for any task $M \in \mathcal{M}_{lin}^\phi$. This implies that $\mathrm{CCS} \subseteq \Psi^{OK}(\Pi_k)$; i.e., the OK matches or exceeds the expressiveness of a CCS.*

*Proof.* We start the proof by stating two assumptions on which the theorem relies:

**Assumption 1.** $\mathrm{NewPolicy}(\mathbf{w})$ returns an optimal policy, $\pi_{\mathbf{w}}^*$, for the given task $\mathbf{w}$.

**Assumption 2.** For all $\mathbf{w} \in \mathcal{W}^{\mathrm{sup}}$, if an optimal policy $\pi_{\mathbf{w}}^* \in \Pi^{\mathrm{OK}}(\Pi)$, then $\mathrm{TrainOK}(\omega, \mathcal{W}^{\mathrm{sup}}, \Pi)$ (Alg. 3) returns a meta-policy, $\omega$, s.t. $\pi_\omega^{\mathrm{OK}}(\cdot, \mathbf{w}; \Pi)$ is optimal w.r.t. task $\mathbf{w}$.

Recall that the OKB (Alg. 1) maintains two partial CCSs, $\Psi_k^{\mathrm{base}}$ and $\Psi_k^{\mathrm{OK}}$, which store, respectively, the SFs of the base policies $\Pi_k$ and the policies generated via the OK using $\Pi_k$. We first will prove that, if in any given iteration $k$, the set of candidate corner weights $\mathcal{C}$ is empty (see line 16 in Alg. 1), then it holds that $\mathrm{CCS} \subseteq \Psi_{k+1}^{\mathrm{OK}}$. First, note that $\mathcal{C}$ contains corner weights of both $\Psi_k^{\mathrm{base}}$ and $\Psi_k^{\mathrm{OK}}$. If the OK policy $\pi_\omega^{\mathrm{OK}}(\cdot; \Pi_k)$ is able to optimally solve all tasks in $\mathcal{C}$ (line 12 in Alg. 1), then we have that $\mathrm{CCS} \in \Psi^{\mathrm{OK}}(\Pi_k)$ due to Lemma A.1. Note that assumptions 1 and 2, above, are necessary because Lemma A.1 requires the partial CCS $\Psi_k$ to be a subset of the CCS. If it contains $\epsilon$-optimal policies, then it is possible to prove convergence to $\epsilon$-CCSs instead (Alegre et al., 2023b).

Now, to conclude the proof, we only need to show that $|\Pi_k| \leq |\mathrm{CCS}|$. The set of base policies is never larger than the CCS due to line 24 in Alg. 1, in which dominated base policies (i.e., redundant policies that are not exclusively optimal w.r.t. any $\mathbf{w}$) are removed from $\Pi_k$ by the procedure $\mathrm{RemoveDominated}$. Importantly, the set of base policies returned by OKB, $\Pi_k$, will only have the same of the CCS ($|\Pi_k| = |\mathrm{CCS}|$) in domains where no single task in $\mathcal{M}_{lin}^\phi$ can be optimally solved by combining policies optimal for other tasks in $\mathcal{M}_{lin}^\phi$.[7] $\qquad\square$

### A.4 Proof of Proposition 4.4

**Proposition** (4.4). *Let $\Pi_k$ be the set of policies learned by OKB (Alg. 1) when solving tasks that are linearly expressible in terms of the reward features $\phi(s, a) \in \mathbb{R}^d$. Let $r$ be an arbitrary reward function that is non-linear with respect to $\phi$. Suppose the optimal policy for $r$ can be expressed by alternating, as a function of the state, between policies in the CCS induced by $\phi$. That is, suppose that for all $s \in \mathcal{S}$, there exists a policy $\pi_{\mathbf{w}}^*$ (optimal for some $\mathbf{w} \in \mathcal{W}$) such that $\pi_r^*(s) = \pi_{\mathbf{w}}^*(s)$. Then, the OK can represent the optimal policy for $r$ using the set of base policies $\Pi_k$; i.e., $\pi_r^* \in \Pi^{\mathrm{OK}}(\Pi_k)$.*

*Proof.* We have that, for all $s \in \mathcal{S}$, $\exists \mathbf{w} \in \mathcal{W}$ such that:

$$\pi_r^*(s) = \pi_{\mathbf{w}}^*(s), \quad \text{where } \boldsymbol{\psi}^{\pi_{\mathbf{w}}^*} \in \mathrm{CCS}, \tag{14}$$

$$= \arg\max_{a \in \mathcal{A}} \boldsymbol{\psi}^{\pi_{\mathbf{w}}^*}(s, a) \cdot \mathbf{w}. \tag{15}$$

Due to Thm. 4.3, we know that by employing the set of base policies returned by OKB, we have that $\pi_{\mathbf{w}}^* \in \Pi^{\mathrm{OK}}(\Pi_k)$. Equivalently,

$$\pi_{\mathbf{w}}^*(s) = \pi_\omega^{\mathrm{OK}}(s, \mathbf{w}; \Pi_k) = \arg\max_{a \in \mathcal{A}} \max_{\pi \in \Pi_k} \boldsymbol{\psi}^\pi(s, a) \cdot \omega(s, \mathbf{w}). \tag{16}$$

---

[7]An example of such a domain is an MDP composed of independent corridors, in which optimal policies for different weights $\mathbf{w} \in \mathcal{W}$ must traverse different corridors.

Combining Eq. (15) and Eq. (16), we have that

$$\pi_r^*(s) = \pi_\omega^{\text{OK}}(s, \mathbf{w}; \Pi_k), \tag{17}$$

which concludes the proof, demonstrating that $\pi_r^* \in \Pi^{\text{OK}}(\Pi_k)$. $\qquad\square$

## A.5 Conditions for OKB's Behavior Basis to be Strictly Smaller Than the CCS

Transfer learning techniques dating back to the early 1990s often leverage the principles of compositionality and modularity—the idea that related tasks can be solved by reusing parts of previously learned solutions.[8] Our formal analysis of when OKB's behavior basis is strictly smaller than the CCS draws on the principles of compositionality and modularity—the idea that related tasks can be solved by combining reusable components ("sub-behaviors") from previously acquired policies. We show that identifying these conditions reduces to analyzing the cardinality of a specific Set Cover problem.

Intuitively, the OKB only requires a behavior basis as large as the CCS in degenerate cases where the solution to *all* possible tasks involves executing at least one type of behavior that does not occur in the solutions to *any* other tasks. Such cases are fundamentally at odds with the assumptions that make transfer learning viable. In contrast, as long as policies in the CCS (which solve related tasks drawn from a shared distribution) produce similar behaviors in shared regions of the state space, the OKB matches the expressiveness of the CCS while requiring strictly fewer policies.

Consider a transfer learning scenario where agents must solve tasks drawn from some distribution, and each task's solution can be expressed as a (possibly stochastic, closed-loop) sequence of sub-behaviors—often referred to as options, skills, policy modules, controllers, motor primitives, or sub-policies. For example, task $\tau_1$ might require sub-behaviors $o_1$, $o_2$, $o_5$, and $o_6$ (opening doors, driving, turning off lights, and finding exits), while task $\tau_2$ might require a different combination of sub-behaviors—some of which it may share with $\tau_1$, since both come from the same distribution.

Assume a distribution of related tasks that induce a CCS whose policies use some subset of the available sub-behaviors, $O = \{o_1, \ldots, o_N\}$, with $\pi_i$ drawing on $O_i \subseteq O$. The question is: What is the smallest subset of CCS policies that together cover all sub-behaviors in $O$? Formally, we seek the smallest collection $\{\pi_j, \pi_k, \pi_r, \ldots\}$ such that $O_j \cup O_k \cup O_r \cup \cdots = O$. This reduces to the classic Set Cover problem and corresponds to identifying the minimum number of CCS policies containing all sub-behaviors that might be required to solve any tasks from the distribution of interest.

Let $\text{SC} \subseteq \text{CCS}$ denote the solution to this Set Cover problem. By Proposition 4.4, if the solution to a novel task consists of switching between policies in a behavior basis, then a corresponding optimal meta-policy exists that sequences such policies. To formally characterize when the OKB is as expressive as a CCS while relying on a strictly smaller behavior basis, we need to understand when $|\text{SC}| < |\text{CCS}|$.

In general, a Set Cover solution will have $|\text{SC}| = |\text{CCS}|$ only in degenerate cases where no CCS policy can be discarded without leaving some sub-behavior uncovered. One such case arises when every policy in the CCS contains at least one unique sub-behavior—a behavior that *never* contributes to solving any other possible task. This situation is arguably at odds with the assumptions of transfer learning. Only in such cases—for example, when every possible task uniquely requires a specialized sub-behavior that no other task ever uses—would $|\text{SC}| = |\text{CCS}|$.

As long as some sub-behaviors are shared between elements of the CCS, the set of policies needed to cover all sub-behaviors (i.e., $|\text{SC}|$) will be strictly smaller than the CCS. In such common transfer learning settings, the OKB is as expressive as a CCS while requiring strictly fewer policies.

## B  Transforming General Linear Reward MDPs into Convex Reward MDPs

Let $\mathcal{M}_{\text{lin}}^\phi \subseteq \mathcal{M}$ be the (possibly infinite) set of MDPs associated with all linearly expressible reward functions. This set, commonly studied in the SFs literature, can be defined as

$$\mathcal{M}_{\text{lin}}^\phi \triangleq \{(\mathcal{S}, \mathcal{A}, p, r_{\mathbf{w}}, \mu, \gamma) \mid r_{\mathbf{w}} = \phi \cdot \mathbf{w}\},$$

---

[8]These principles underpin a long line of research on hierarchical reinforcement learning, temporal abstraction, macro-actions, skill discovery, and studies in neuroscience and cognitive science exploring how animal behavior is organized hierarchically.

where $\mathbf{w} \in \mathbb{R}^d$ is a $d$-dimensional weight vector that may include negative elements.

Standard methods for computing convex coverage sets (CCS) require reward functions to be expressed as *convex* combinations of features, i.e., using weights in the $d$-dimensional simplex, $\Delta^d$:

$$\Delta^d = \left\{ \mathbf{w} \in \mathbb{R}^d \mid w_i \geq 0, \sum_{i=1}^d w_i = 1 \right\}.$$

To use methods that learn a CCS—typically designed for convex reward functions—to solve all *linear* tasks in $\mathcal{M}_{\text{lin}}^\phi$, we first need to show that any MDP in $\mathcal{M}_{\text{lin}}^\phi$ can be transformed into an equivalent one where rewards are expressed as convex combinations over a new set of features. In other words, given a family of MDPs $\mathcal{M}_{\text{lin}}^\phi$, we show how to construct a corresponding family of MDPs with *convex* rewards (defined over a transformed feature space $\tilde{\phi}$) that induces the same set of optimal policies. This family of MDPs can be defined as follows:

$$\mathcal{M}_{conv}^{\tilde{\phi}} \triangleq \{(\mathcal{S}, \mathcal{A}, p, r_{\mathbf{w}}, \mu, \gamma) \mid r_{\mathbf{w}} = \tilde{\phi} \cdot \mathbf{w}, \mathbf{w} \in \Delta^{2d}\}, \text{ where } \tilde{\phi}(s,a) = \begin{bmatrix} \phi(s,a) \\ -\phi(s,a) \end{bmatrix} \in \mathbb{R}^{2d}. \tag{18}$$

We now show that this transformed family of MDPs induces the same set of optimal policies as $\mathcal{M}_{\text{lin}}^\phi$. As a result, learning a CCS for $\mathcal{M}_{conv}^{\tilde{\phi}}$ allows us to recover the optimal policy for any task in $\mathcal{M}_{\text{lin}}^\phi$. Let $r(s,a) = \phi(s,a) \cdot \mathbf{w}$ be an arbitrary linear reward function with a weight vector $\mathbf{w} \in \mathbb{R}^d$. One can decompose $\mathbf{w}$ into its non-negative and non-positive components as follows:

$$\mathbf{w} = \mathbf{w}^+ - \mathbf{w}^-, \quad \text{where} \quad w_i^+ = \max(w_i, 0), \quad w_i^- = \max(-w_i, 0).$$

We can now define an augmented feature vector $\tilde{\phi}$:

$$\tilde{\phi}(s,a) = \begin{bmatrix} \phi(s,a) \\ -\phi(s,a) \end{bmatrix} \in \mathbb{R}^{2d},$$

and a non-negative weight vector $\tilde{\mathbf{w}}$:

$$\tilde{\mathbf{w}} = \begin{bmatrix} \mathbf{w}^+ \\ \mathbf{w}^- \end{bmatrix} \in \mathbb{R}_{\geq 0}^{2d}.$$

This allows us to rewrite the original reward function as follows:

$$r(s,a) = \mathbf{w} \cdot \phi(s,a) = \tilde{\mathbf{w}} \cdot \tilde{\phi}(s,a).$$

Finally, let us normalize the weights $\tilde{\mathbf{w}}$ (e.g., by dividing by their norm) to ensure they lie in the simplex $\Delta^{2d}$. Recall that scaling a reward function by a positive constant does not affect its optimal policy, so this transformation is valid. Since we showed that any linear reward function can be expressed as a convex combination of a transformed feature set—without altering the corresponding optimal policy—it follows that a CCS for $\mathcal{M}_{conv}^{\tilde{\phi}}$ contains the optimal solutions for all tasks in the original family, $\mathcal{M}_{\text{lin}}^\phi$.

## C Beyond Linear Rewards

In this section, we further discuss the use of the OK to solve tasks defined by non-linear reward functions; that is, reward functions that are not linear-expressible under reward features $\phi(s, a, s') \in \mathbb{R}^d$.

We start by arguing that, in many cases, the linear reward assumption does not pose any limitations. As discussed by Barreto et al. (2020), when both $\mathcal{S}$ and $\mathcal{A}$ are finite, we can recover any possible reward function by defining $d = |\mathcal{S}|^2 \times |\mathcal{A}|$ features $\phi_i$, each being an indicator function associated with a specific transition $(s, a, s')$. This result shows that it is possible to define reward features $\phi \in \mathbb{R}^{|\mathcal{S}|^2 \times |\mathcal{A}|}$, such that every task in $\mathcal{M}$ is linearly expressible. A challenge, however, is to find alternative features with this property, but with dimension $d \ll |\mathcal{S}|$ (Touati and Ollivier, 2021).

When examining how the OK is defined (Eq. 4), a natural idea is to use it to solve tasks defined by reward functions with state-dependent weights $\mathbf{w}(s)$. In particular, let $\mathcal{M}^\phi_{\text{sd-linear}}$ be a family of non-linear tasks defined as

$$\mathcal{M}^\phi_{\text{sd-linear}} \triangleq \{(\mathcal{S}, \mathcal{A}, p, r_{\mathbf{w}}, \mu, \gamma) \mid r_{\mathbf{w}}(s, a, s') = \phi(s, a, s') \cdot \mathbf{w}(s)\}. \tag{19}$$

Note that it is easy to show that $\mathcal{M}^\phi_{\text{lin}} \subseteq \mathcal{M}^\phi_{\text{sd-linear}} \subseteq \mathcal{M}$.

Unfortunately, the OK may be unable to solve all tasks in the family $\mathcal{M}^\phi_{\text{sd-linear}}$, even when given access to a set of policies $\Pi_k$ forming a CCS.

**Proposition C.1.** *There exists a task $M \in \mathcal{M}^\phi_{\text{sd-linear}}$ such that no meta-policy, $\omega : \mathcal{S} \to \mathcal{Z}$, and set of policies, $\Pi$, results in $\pi^{OK}_\omega(s; \Pi)$ being optimal with respect to $M$.*

*Proof.* Assume a family of MDPs, $M \in \mathcal{M}^\phi_{\text{sd-linear}}$, with states, actions, transitions, and features defined as in Figure 6. Let a task $M \in \mathcal{M}^\phi_{\text{sd-linear}}$ be defined via the following reward function:

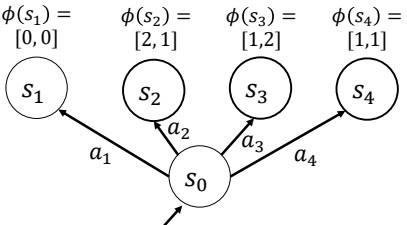

Figure 6: Counterexample MDP.

$r(s_1) = \phi(s_1) \cdot [1, 0] = 0, r(s_2) = \phi(s_2) \cdot [-1, -1] = -3, r(s_3) = \phi(s_3) \cdot [-1, -1] = -3, r(s_4) = \phi(s_4) \cdot [1, 1] = 2$, with optimal policy $\pi^*(s_0) = a_4$ and whose SF vector is $\psi^{\pi^*} = [1, 1]$. We now show that there is no meta-policy $\omega(s)$ that results in $\pi^{\text{GPI}}(s; \omega(s))$ being equal to the optimal policy $\pi^*$. This is proved by noticing the impossibility of the following equality:

$$\pi^{\text{GPI}}(s_0, \omega(s_0)) = \arg\max_{a \in \{a_1, a_2, a_3, a_4\}} \max_{\pi \in \Pi} \psi^\pi(s_0, a) \cdot \omega(s_0) = a_4. \tag{20}$$

$\square$

Notice that the result above implies that there are tasks in $\mathcal{M}^\phi_{\text{sd-linear}}$ that can not be solved by $\pi^{\text{OK}}$ even when assuming access to a CCS.

While this result may be negative, we highlight that $\mathcal{M}^\phi_{\text{sd-linear}}$ is a family of tasks more general than it may seem at first glance.

**Proposition C.2.** *Let $\mathcal{M}' \subset \mathcal{M}$ such that for all $M \in \mathcal{M}'$, $r(s, a, s') = r(s)$. If $\phi(s) \neq \mathbf{0}$ for all $s \in \mathcal{S}$, then $\mathcal{M}^\phi_{\text{sd-linear}} = \mathcal{M}'$.*

*Proof.* If $\phi(s) \neq \mathbf{0}$ for all $s \in \mathcal{S}$, then for any reward function $r$ there exist a function $\mathbf{w}(s)$ such that $r(s) = \phi(s) \cdot \mathbf{w}(s)$. $\square$

The proposition above implies that, under mild conditions, $\mathcal{M}^\phi_{\text{sd-linear}}$ is able to represent *all* reward functions and tasks in $\mathcal{M}$.

## D   Training the OK Meta-Policy

Alg. 3 (Train-OK) is the sub-routine for training a meta-policy omega under an actor-critic training framework, given the selected tasks by Alg. 2. In Alg. 3, we show how our method optimizes the OKB meta-policy $\omega$ for a given set of tasks $\mathcal{W}^{\text{sup}}$.

**Algorithm 3:** Train Option Keyboard (`TrainOK`)

---

**Input:** Meta-policy $\omega$, weight support $\mathcal{W}^{\text{sup}}$, base policies $\Pi$.

1   Initialize replay buffer $\mathcal{B}$
2   Let $\psi^\omega(s, \mathbf{z}, \mathbf{w})$ be the critic of the meta-policy $\omega$
3   $\mathbf{w}_0 \sim \mathcal{W}^{\text{sup}}$; $S_0 \sim \mu$
4   **for** $t = 0, 1, 2, \dots, T$ **do**
5      **if** $S_t$ *is terminal* **then**
6         $\mathbf{w}_t \sim \mathcal{W}^{\text{sup}}$
7         $S_t \sim \mu$
8      $\mathbf{z}_t \leftarrow \omega(S_t, \mathbf{w})$
      ▷ Exploration clipped Gaussian noise, as in TD3
9      $\mathbf{z}_t \leftarrow \mathbf{z}_t + \text{clip}(\epsilon, -0.5, 0.5)$, where $\epsilon \sim \mathcal{N}(0, 0.2^2)$
10     $\mathbf{z}_t \leftarrow \mathbf{z}_t / \|\mathbf{z}_t\|_2$
      ▷ Follow OK policy
11     $A_t \leftarrow \pi^{\text{GPI}}(S_t, \mathbf{z}_t; \Pi)$
12     Execute $A_t$, observe $S_{t+1}$, and $\boldsymbol{\phi}_t$
13     Add $(S_t, \mathbf{z}_t, \boldsymbol{\phi}_t, S_{t+1})$ to $\mathcal{B}$
14     Update $\psi^\omega$ by minimizing

$$\mathbb{E}_{(s,\mathbf{z},\phi,s') \sim \mathcal{B}, \mathbf{w} \sim \mathcal{W}^{\text{sup}}} \left[ \left( \psi^\omega(s, \mathbf{z}, \mathbf{w}) - (\boldsymbol{\phi} + \gamma \psi^\omega(s', \omega(s', \mathbf{w}), \mathbf{w})) \right)^2 \right]$$

15     Update $\omega$ via the policy gradient $\nabla_{\mathbf{z}} \psi^\omega(s, \mathbf{z}, \mathbf{w}) \cdot \mathbf{w}|_{\mathbf{z}=\omega(s,\mathbf{w})} \nabla_\omega \omega(s, \mathbf{w})$
16   **return** $\omega$

---

## E   Checking the Condition for OK Optimality

Our practical implementation of the OKB test the condition in Thm. 4.2 by randomly sampling from a replay buffer, $\mathcal{B} = \{(s_i, a_i, \phi(s_i, a_i), s_i')\}_{i=1}^n$, which contains experiences observed during the training of the base policies in $\Pi_k$. Given a candidate task $\mathbf{w} \in \mathcal{C}$ (line 12 of Alg. 1), we compute the *mean positive advantage* of $\omega$ as:

$$\left( \sum_{i=1}^{|\mathcal{B}|} \mathbb{1}_{\{\hat{A}_{\mathbf{w}}^\omega(s_i, a_i) > 0\}} \right)^{-1} \sum_{i=1}^{|\mathcal{B}|} \max(\hat{A}_{\mathbf{w}}^\omega(s_i, a_i), 0), \tag{21}$$

where $\hat{A}_{\mathbf{w}}^\omega(s_i, a_i) = \phi(s_i, a_i) \cdot \mathbf{w} + \gamma \psi^\omega(s_i', \omega(s_i')) \cdot \mathbf{w} - \psi^\omega(s_i, \omega(s_i)) \cdot \mathbf{w}$. In line 14 of Alg. 1, we select the task $\mathbf{w} \in \mathcal{C}$ with the highest value for Eq. (21). Intuitively, we select the task $\mathbf{w}$ for which the OK can improve the most its performance. By adding $\pi_{\mathbf{w}}$ to $\Pi_k$ and retraining $\omega$, the OK will be more expressive in the subsequent iteration. We highlight that OKB theoretical guarantees (e.g., Thm. 4.3) are independent of the heuristic used to select the task $\mathbf{w} \in \mathcal{C}$ (line 14), and other strategies could be used instead.

## F   Experimental Details

The code and scripts necessary to reproduce our experiments will be made publicly available upon acceptance.

The USFAs $\psi(s, a, \mathbf{w})$ used for encoding the base policies $\Pi_k$ were modeled with multi-layer perceptron (MLP) neural networks with 4 layers with 256 neurons. We use an ensemble of 10 neural networks, similar to Chen et al. (2021), and compute the minimum value over two randomly sampled elements when computing the Bellman update targets. We used Leaky ReLU activation functions and layer normalization for improved training stability. For a fair comparison, we independently optimized all method-specific hyperparameters (e.g., network architectures, OKB settings, and those of competing baselines) via grid search.

The budget of environment interactions per iteration (i.e., call to `NewPolicy` in Alg. 1) used was 25000, 50000, 50000 and 100000 for the Minecart, FetchPickAndPlace, Item Collection, and High-way domains, respectively. At each iteration, $\psi(s, a, \mathbf{w})$ is trained with the current task $\mathbf{w}$, as well as with the tasks from previous iterations in order to avoid catastrophic forgetting.

The meta-policy $\omega(s, \mathbf{w})$ was modeled with an MLP with 3 layers with 256 neurons. We employed the techniques introduced by Bhatt et al. (2024), i.e., batch normalization and removal of target networks, which increased the training efficiency. We used Adam (Kingma and Ba, 2015) as the first-order optimizer used to train all neural networks with mini-batches of size 256.

When training the base policies, we used $\varepsilon$-greedy exploration with a linearly decaying schedule. For training the meta-policy, we added a clipped Gaussian noise (see line 11 of Alg. 3), as done in other actor-critic algorithms, e.g., TD3 (Fujimoto et al., 2018).

Since running OK-LS (Alg. 2) until no more corner weights are identified (see line 4) may require a large number of iterations, in the experiments we ran OK-LS for 5 iterations, which we found to be enough for learning a well-performing meta-policy $\omega$.

To generate sets of test tasks $\mathcal{W}' \subset \mathcal{W}$ given different values of $d$, we employed the method introduced by Takagi et al. (2020) available on pymoo (Blank and Deb, 2020), which produces uniformly-spaced weight vectors in the simplex, $\mathcal{W}$.

All experiments were performed in a cluster with NVIDIA A100-PCIE-40GB GPUs with 32GB of RAM. Each individual run of each method took approximately 2.5 hours (Item Collection), 6 hours (Minecart), 10 hours (FetchPickAndPlace), and 25 hours (Highway).

## F.1 Corner Weights.

Below, we define the concept of corner weights used by OKB (Alg. 1) and OK-LS (Alg. 2), as defined in previous works (Roijers, 2016; Alegre et al., 2022).

**Definition F.1.** Let $\Psi = \{\boldsymbol{\psi}^{\pi_i}\}_{i=1}^{n}$ be a set of SF vectors of $n$ policies. *Corner weights* are the weights contained in the vertices of a polyhedron, $P$, defined as:

$$P = \{\mathbf{x} \in \mathbb{R}^{d+1} \mid \mathbf{V}^+\mathbf{x} \leq \mathbf{0}, \sum_i w_i = 1, w_i \geq 0\}, \tag{22}$$

where $\mathbf{V}^+$ is a matrix whose rows store the elements of $\Psi$ and is augmented by a column vector of $-1$'s. Each vector $\mathbf{x}=(w_1, ..., w_d, v_{\mathbf{w}})$ in $P$ is composed of a weight vector and its scalarized value.

To compute corner weights, as in Def. F.1, we used pycddlib (`https://github.com/mcmtroffaes/pycddlib`) implementation of the Double Description Method (Motzkin et al., 1953) to efficiently enumerate the vertices of the polyhedron $P$.

## F.2 Domains

In this section, we describe in detail the domains used in the experiments, which are shown in Fig. 1.

**Minecart domain.** The Minecart domain is a widely-used benchmark in the multi-objective reinforcement learning literature (Abels et al., 2019). We used the implementation available on MO-Gymnasium (Felten et al., 2023). This domain consists of a cart that must collect two different ores and return them to the base while minimizing fuel consumption. The agent' observation $\mathcal{S} \subset \mathbb{R}^7$ contains the agent $x, y$ position, the current speed of the cart, its orientation ($\sin$ and $\cos$), and the percentage of occupied capacity in the cart by each ore: $\mathcal{S} = [-1, 1]^5 \times [0, 1]^2$. The agent has the choice between 6 actions: $\mathcal{A} = \{\text{MINE, LEFT, RIGHT, ACCELERATE, BRAKE, DO NOTHING}\}$. Each mine has a different distribution over two types of ores. Fuel is consumed at every time step, and extra fuel is consumed when the agent accelerates or selects the mine action. The reward features of this domain, $\phi(s, a, s') \in \mathbb{R}^3$, is defined as:

$$\phi_1(s, a, s') = \text{quantity of ore 1 collected if } s' \text{ is inside the base, else } 0,$$
$$\phi_2(s, a, s') = \text{quantity of ore 2 collected if } s' \text{ is inside the base, else } 0,$$
$$\phi_3(s, a, s') = -0.005 - 0.025\mathbb{1}\{a = \text{ACCELERATE}\} - 0.05\mathbb{1}\{a = \text{MiNE}\}.$$

We used $\gamma = 0.98$ in this domain.

**FetchPickAndPlace domain.** We extended the FetchPickAndPlace domain (Plappert et al., 2018), which consists of a fetch robotic arm that must grab a block on the table with its gripper and move the block to a given target position on top of the table (shown in the middle of Fig. 1). Our implementation

of this domain is an adaptation of the one available in Gymnasium-Robotics (de Lazcano et al., 2023). We note that the state space of this domain, $\mathcal{S} \subset \mathbb{R}^{25}$, is high-dimensional. The action space consists of discretized Manhattan-style movements for the three movement axes, i.e., $\{1 : [1.0, 0.0, 0.0], 2 : [-1.0, 0.0, 0.0], 3 : [0.0, 1.0, 0.0], 4 : [0.0, -1.0, 0.0], 5 : [0.0, 0.0, 1.0], 6 : [0.0, 0.0, -1.0]\}$, and two actions for opening and closing the gripper, totaling 8 actions. The reward features $\phi(s, a, s') \in \mathbb{R}^d$ correspond to the negative Euclidean distances between the square block and the $d$ target locations (shown in red in Fig. 1). We used $\gamma = 0.95$ in this domain.

**Item Collection domain.** This domain consists of a $10 \times 10$ grid world, in which an agent (depicted as a yellow circle in the rightmost panel of Fig. 1) moves along the 4 directions, $\mathcal{A} = \{\text{UP}, \text{DOWN}, \text{LEFT}, \text{RIGHT}\}$, and must collect items of two different types (denoted by red triangles and green diamonds, respectively). The reward features $\phi(s, a, s') \in \mathbb{R}^2$ are indicator functions indicating whether the agent collected one of the items in state $s'$. In each episode, 5 items of each type are randomly placed in the grid. The agent perceives its observation as a $10 \times 10$ image with 2 channels (one per item), which is flattened into a 200-dimensional vector. We make the observations and dynamics toroidal—that is, the grid "wraps around" connecting cells on opposite edges—similarly as done by Barreto et al. (2019). We used $\gamma = 0.95$ in this domain.

**Highway domain.** This domain is based on the autonomous driving environment introduced by Leurent (2018). The agent controls a vehicle on a multilane highway populated with other vehicles. Its observations includes its coordinates as well as coordinates from other vehicles. Formally, the state space is given by an array of size $V \times F$, where $V$ represents the number of vehicles in the environment and $F$ are the features describing each vehicle, e.g., velocity, position, angle. The agent's actions consist in changing lanes, accelerating or decelerating, or doing nothing, i.e., $\mathcal{A} = \{\text{TURN\_LEFT}, \text{IDLE}, \text{TURN\_RIGHT}, \text{FASTER}, \text{SLOWER}\}$. The reward features of this domain, $\phi(s, a, s') \in \mathbb{R}^3$, is defined as:

$$\phi_1(s, a, s') = \text{normalized forward speed of the vehicle,}$$
$$\phi_2(s, a, s') = 0.5 \text{ if driving in the rightmost lane, else } 0,$$
$$\phi_3(s, a, s') = -1 \text{ if } a = \text{TURN\_LEFT or } a = \text{TURN\_RIGHT else } 0.$$

All three reward features above are zeroed if the agent is off the road and penalized by $-10$ if the agent crashes into another vehicle. We used $\gamma = 0.99$ in this domain.

