# OpenReview forum: "Constructing an Optimal Behavior Basis for the Option Keyboard"
_NeurIPS.cc/2025/Conference — NeurIPS 2025 poster_

### Official Review · Reviewer_g75S · 2025-06-07

**Clarity:** 3
**Significance:** 3
**Originality:** 3
**Rating:** 5
**Confidence:** 1

**Summary:**

This paper is about a method for efficiently constructing an optimal behavior basis (a set of so-called base policies) for solving multi-task reinforcement learning. Based this optimal behavior basis a dynamic combination of these policies is used to solve new RL tasks. The underlying concept that make this work are successor features. The paper provides strong formal guarantees that this optimal basis allows the options keyboard (a generalization of GPI with different weights in each state) to optimally solve any new linear task. One of the theoretical contributions is a proof that this can express a strictly larger set of policies than the convex combination baseline. The approach is evaluated on high-dimensional RL problems and shows favorable performance relative to baselines.

**Questions:**

- See weaknesses.

- How do you designed suitable successor features?

**Ethical Concerns:**

["NO or VERY MINOR ethics concerns only"]

**Final Justification:**

I thank the authors for their responses (also regarding the comments of the other reviews). At this point, I do not have any further questions and I remain positive about this paper and maintain my ratings.

**Limitations:**

yes

**Paper Formatting Concerns:**

-

**Quality:**

3

**Strengths And Weaknesses:**

Strengths
- The paper is well-written and addresses an important and interesting problem.
- From what I understand, the paper is correct and provides a novel and interesting solution. However, I am not an expert in this area and cannot claim to understand the theoretical results of the paper completely.
- The research questions are clearly phrased in the experiments section which helps understanding the evaluation.
- Given the results, the method seems to extend to non-linear reward functions to a certain degree.
- Evaluation is done with 15 seeds.
- The method is shown to significantly reduce the number of needed base policies which is very relevant.
- I am sure the paper is of high quality and I would like to see an expert reviewer make a statement about the theoretical contributions which are extensive (considering the appendix).

Weaknesses
- I a bit puzzled by the question of when such an approach is actually practical. The assumption that this approach makes seem challenging for a real-world RL tasks / settings. Perhaps, the authors can argue more for practical suitability beyond the simulated robot example.

---

> ### Author Rebuttal · Authors · 2025-07-31
>
> We thank the reviewer for the positive and encouraging feedback. We are glad you found the paper to be clear, novel, and impactful. We also appreciate your comments on the correctness of our approach, the strength of our empirical evaluation, and the significance of our method’s ability to produce an optimal behavior basis significantly smaller than a CCS: this was precisely the central goal of the work.
>
> Below, we address the reviewer’s questions and comments:
>
> ---
>
> ### Questions regarding the practical applicability of our method
>
> The reviewer raised an excellent question about the practical suitability of our approach beyond robotics examples. This is indeed a crucial point, since Successor Features (SFs)—the core mathematical object underlying our method—form the foundation of many recent advances in reinforcement learning.
>
> A key reason SFs scale well to large problems is that computing them reduces to estimating a vectorized value function, which allows direct application of state-of-the-art TD learning algorithms. These techniques have been shown to scale to domains with over a million state features, including computer Go. Accordingly, SF-based methods that build upon such techniques have successfully extended to high-dimensional settings such as 3D image-based navigation and robotic locomotion (Carvalho et al., 2023a; Chua et al., 2024). Our method can directly leverage all of these advances.
>
> Moreover, please note that our approach is the first to ensure zero-shot optimality without requiring explicit construction of exponentially large Convex Coverage Sets (CCS), a key limitation in existing methods. This results in a substantial improvement in scalability over prior approaches.
>
> Finally, we would like to highlight that:
>
> **(1)** Our results go beyond demonstrating that OKB consistently outperforms state-of-the-art competitors in the most relevant and challenging benchmarks in the field (e.g., the autonomous driving environment in Figure 5). They also show that our method’s performance advantage over existing techniques becomes *increasingly pronounced* as task complexity increases.
>
> **(2)** Prior methods have only been evaluated on problems with dimensionality $d \leq 4$ (e.g., in the original SFOLS and SIP papers). Our paper is the first to demonstrate effective, scalable, and provably optimal zero-shot transfer in problems with twice as many dimensions. This shows not only that our technique outperforms all relevant baselines, but also that it does so in problems an order of magnitude more complex than those addressed in prior work.
>
> This is a great question and helped us realize the need to better emphasize that the core components of our approach (SFs) have already been systematically shown to scale to problems with millions of features, and that our own results provide the first empirical evidence of consistent outperformance over state-of-the-art competitors on domains with *twice* the dimensionality of those studied in prior work.
>
> ---
>
> ### Designing Successor Features
>
> The paper details how the reward features $\phi$ are designed for each domain in Appendix F.2. For example, in the Minecart domain, $\phi$ includes quantities of collected ores and fuel consumption. In Fetch-PickAndPlace, $\phi$ represents negative Euclidean distances to target locations. For Item Collection, $\phi$ is composed of indicator functions for collected items, and in Highway, $\phi$ includes normalized forward speed, lane position, and turning actions. These are hand-crafted features based on domain knowledge, which is a common practice in the SF literature.
> Details about the architecture and training of the neural networks used to model the SFs follow standard practices in deep RL and are described in Appendix F.
>
> ---
>
> We thank the reviewer again for the positive and supportive review. Given your primary questions regarding scalability of the approach, we hope to have clarified the practical applicability of our method and SFs in general.
> We note that the reviewer said in their review that they are “sure the paper is of high quality” and they would like to see an expert reviewer make a statement about the theoretical contributions.
> Other reviewers also pointed out that: “OKB produces a set of base policies and meta-policy which are optimal with respect to the family of tasks while requiring fewer base policies than existing methods” (Reviewer rDD6); “the proposed method is novel and theoretically sound” (Reviewer 1Yjh); “the policy basis is indeed smaller than that in previous work that also guarantee optimality” (Reviewer QsaG).
>
> If you believe the clarifications above address the questions raised, and in light of the positive comments of the other reviewers regarding our theoretical contributions, we would be grateful if you would consider revisiting your score and confidence level. If any points remain unclear, we welcome further feedback and would be happy to continue the discussion. Thank you again for your thoughtful comments!

---

> > ### Comment · Reviewer_g75S · 2025-08-05
> >
> > I thank the authors for their responses (also regarding the comments of the other reviews). I have a better understanding of the scalability of the method now. At this point, I do not have any further questions and I remain positive about this paper and maintain my ratings.

---

> > > ### Author Response · Authors · 2025-08-08
> > >
> > > We thank the reviewer once again for the thoughtful discussion, constructive feedback, and valuable comments. We greatly appreciate their positive and encouraging remarks on the relevance and importance of our contributions, as well as their support for the acceptance of our paper.

---

### Official Review · Reviewer_QsaG · 2025-07-01

**Clarity:** 3
**Significance:** 2
**Originality:** 3
**Rating:** 3
**Confidence:** 4

**Summary:**

The paper proposes a novel approach to multitask reinforcement learning with successor features. Concretely, the contribution consists in computing a policy basis that enables zero-shot identification of an optimal policy for any linear task. The policy basis can be significantly smaller than a convex coverage set, and generalization occurs by considering linear combinations of the policies in the basis. The contribution is empirically tested in a set of benchmark domains.

**Questions:**

Proposition 4.4 states that the OK *can* represent an optimal policy for a non-linear reward function. However, how would you actually compute the function \omega(s,w) in this setting, if there is no linear task w?

In the experiments, the authors use a USFA parameterized on the task w to learn the successor features. This seems like a peculiar choice given the problem at hand. On one hand, this will likely introduce approximation errors. On the other hand, this allows representing the successor features of *any* policy, not just those in a small set.

There seems to be a typo in Theorem 4.2: in A^{w^*}r(s,a), r should supposedly be a quantifier of A.

**Ethical Concerns:**

["NO or VERY MINOR ethics concerns only"]

**Final Justification:**

After discussion with the authors I have raised my score, since some of my concerns were clarified (specifically, the number of successor feature vectors computed by the algorithm). I believe that the final version of the paper needs to be more honest regarding the contribution, and include the theoretical justification provided by the authors in the discussion.

**Limitations:**

Notably, a discussion around the number of successor feature vectors computed by the algorithm is lacking.

**Paper Formatting Concerns:**

No concerns.

**Quality:**

2

**Strengths And Weaknesses:**

The paper is clearly written, and achieving efficient zero-shot identification of an optimal policy in a family of MDPs is an important research question. It is also true that the policy basis is indeed smaller than that in previous work that also guarantee optimality.

On the negative side, I am not convinced that the proposed algorithm leads to more efficient learning than previous work. The reason is that the computational cost is dominated by the number of successor feature vectors, not the number of base policies. Specifically, learning the successor feature vector of a single policy is as hard as solving a single MDP in the given family. From the description of the algorithm (on page 4 and in Algorithm 1) it appears that the number of successor feature vectors in the set \Psi^{OK} is *larger* than that of a convex coverage set, which would imply that the overall computational complexity is greater. The authors do not discuss why this is the case, but my intuition is that we cannot linearly compose successor feature vectors of different policies, making it necessary to explicitly compute a successor feature vector for each policy considered (including many linear combinations of the base policies).

---

> ### Author Rebuttal · Authors · 2025-07-31
>
> We thank the reviewer for the insightful review. We appreciate their positive comments acknowledging that our method is capable of producing a basis significantly smaller than a CCS while ensuring zero-shot optimality—this was precisely the primary goal of our paper.
>
> Below, we address the reviewer's main concerns and questions:
>
> ---
>
> ### On the dominant computational cost in our approach
>
> Thank you for your questions. Your comments helped us realize that we should more clearly emphasize the distinct roles played by the two sets central to our method—namely, $\Pi_k$ and $\Psi^{OK}$.
>
> The reviewer makes three claims: *(a)* learning an SF vector is as hard as solving an MDP; *(b)* the computational cost of our method is dominated by the number of successor feature (SF) vectors rather than the number of base policies; and *\(c\)* the overall cost of learning $\Psi^{OK}$ exceeds that of learning a CCS.
>
> To carefully address these, we begin with a broader overview of our method and its key properties, followed by a more technical and formal discussion.
>
> ### Overview and definitions
>
> **(1)** Two sets of policies are central to our method—both for formally characterizing its properties and for designing the algorithm itself: $\Pi_k$ and $\Psi^{OK}$. The set $\Pi_k$ is a *behavior basis*—a small, compact set of policies (much smaller than the CCS) that our method combines to solve new tasks (line 153). In contrast, $\Psi^{OK}$ denotes the set of all optimal policies (more precisely, their SF vectors) that can be directly reconstructed by our method by combining the policies in $\Pi_k$ (line 151). These are the strategies our method can deploy to solve novel tasks zero-shot, without the need to train a specialized policy for each task—a process that is often costly, time-consuming, or infeasible.
>
> **(2)** A key property of our method—central to its effectiveness and flexibility—is its provable ability to directly reconstruct all policies in a CCS, as well as policies for a challenging family of non-linear tasks, without ever explicitly having to compute or optimize them. Instead, it is capable of reconstructing them on demand using a meta-policy $\omega$ that combines policies from the behavior basis $\Pi_k$. These formal guarantees are established in Theorem 4.3 and extended by Proposition 4.4.
>
> **(3)** In our paper, we formally express this property by stating that the CCS is a subset of $\Psi^{OK}$; that is, CCS $\subseteq \Psi^{OK}$. Intuitively, this means that our method is not only capable of reconstructing all optimal solutions for linear tasks, but it generalizes beyond the CCS and directly synthesizes optimal policies for particular non-linear tasks.
>
> **(4)** Importantly, note that the formal statement CCS $\subseteq \Psi^{OK}$ is a claim about the ***expressiveness*** of our method—specifically, which optimal policies it is capable of directly reconstructing, zero-shot, without requiring any training. It is ***not*** a statement about which (or how many) policies must be learned to construct the behavior basis that enables such zero-shot capability.
>
> ### Addressing the three primary points raised by the reviewer
>
> Given the overview above, we now respond to the three key claims raised by the reviewer.
>
> ---
> **(a)** ***”Learning an SF vector is as hard as solving an MDP”****.
>
> Unlike what the reviewer claims, learning SF vectors is *not* as hard as solving an MDP.
>
> Estimating the successor features associated with a policy is a *policy evaluation* problem. Solving an MDP, by contrast, is a *policy optimization* problem. The sample complexity of solving an MDP and identifying an $\varepsilon$-optimal policy is approximately $\tilde{O}\Big( \frac{|S| \, |A|}{(1-\gamma)^3 \varepsilon^2} \Big)$. The sample complexity of evaluating a policy (specifically, computing its successor features with respect to the MDP’s initial state, as done in our paper; see line 97) is $\tilde{O}\Big( \frac{1}{(1-\gamma)^2 \varepsilon^2}\Big)$.
>
> These well-known results imply that solving an MDP requires $\tilde{O}\Big( \frac{|S| \, |A|}{(1-\gamma)} \Big)$ more samples than evaluating a fixed policy (e.g., computing its SFs). As $\gamma$ approaches 1 (as is common in the literature), solving an MDP becomes orders of magnitude harder than computing its SFs. As a concrete example, consider that in deterministic environments, evaluating a policy—computing its SFs—can be done by simply executing the policy *once*, over a *single episode*, and recording the resulting rewards. By contrast, learning an optimal policy for standard RL benchmarks often requires thousands or millions of environment interactions.
>
> Hence, we respectfully point out that the reviewer’s claim—that learning SFs is as hard as solving an MDP—is incorrect. This misunderstanding also appears to underlie the two subsequent points they raised; please see below.
>
> ---
> **(b)**  ***“The computational cost of [our method] is dominated by the number of SF vectors, not the number of base policies”***.
>
> We believe this claim stems from the reviewer’s observation that $\Psi^{OK}$ stores SF vectors, along with our statement that CCS $\subseteq \Psi^{OK}$. However, as clarified in point **(4)** above, the cardinality of $\Psi^{OK}$ is not related to the number of base policies (or SFs) the agent must learn, and the notation CCS $\subseteq \Psi^{OK}$ does not imply that our method needs to compute more policies than those in a CCS. Rather, these statements merely indicate that our method is capable of *reconstructing* a larger set of optimal policies than those in a CCS—specifically, by combining policies from the behavior basis $\Pi_k$, where $|\Pi_k| \leq$ $|$CCS$|$ (line 153).
>
> The observation that the size of $\Psi^{OK}$ is unrelated to the number of base policies (or SFs) the agent must learn—combined with the result from point **(a)** above, which states that learning base policies requires orders of magnitude more samples than learning SFs—implies that the computational cost of our method is *not* dominated by the number of SF vectors. Instead, the primary computational bottleneck lies in training the base policies that make up the behavior basis $\Pi_k$.
>
>
> ---
> **(c)** ***The overall cost of learning $\Psi^{OK}$ exceeds that of learning a CCS***.
>
> To address this claim, we first note the set $\Psi^{OK}(\Pi_k)$, as defined in Section 3 line 151, is *never explicitly computed* by Algorithm 1. Instead, as discussed earlier, this set *characterizes* the optimal policies (more precisely, their SF vectors) that can be directly reconstructed by our method by combining the policies in $\Pi_k$.
> The set $\Psi^{OK}$ that appears in Algorithm 1, on the other hand, is updated iteratively and stores the SF vectors of the CCS policies our method becomes capable of reconstructing. Hence, it is *never larger than the CCS* itself. Our algorithm terminates when this set converges to the CCS (line 12), at which point $\Psi^{OK} = CCS$ and $|\Pi_k| \leq |CCS|$.
> Thus, contrary to the reviewer's claim, our method does not have to learn all policies in a CCS, and certainly not more. This is formally established in Theorem 4.3, which is central to our theoretical contributions.
>
> ---
>
> ### Using a USFA to learn SFs
>
> We note that the use of USFAs is common practice in the SF literature—see, e.g., Borsa et al. (2019), Kim et al. (2022), and Carvalho et al. (2023a,b). Our motivation for using a USFA is to avoid the memory and computational costs associated with training a separate neural network for each policy’s SFs. Importantly, a USFA does not necessarily introduce more approximation error than training separate networks—one per policy—as generalization in both cases depends primarily on the network architecture, capacity, and the distribution of training data.
>
> Finally, although a USFA is theoretically capable of approximating the SFs of any policies for linear tasks, in practice we train it (and rely on its predictions) only for the tasks $w$ on which the OKB is trained on. By limiting its use in this way, we reduce potential generalization errors. We agree this is an important point and will clarify it further in the updated version of the paper.
>
>
> ### Proposition 4.4
>
> The reviewer asked how to learn a meta-policy $\omega$ in settings where the OKB is used to reconstruct the optimal policy for non-linear tasks. The reviewer is correct that when generalizing over linear tasks $w$, the meta-policy takes the form $\omega(s, w)$. In contrast, when solving potentially non-linear tasks (as in Proposition 4.4), the meta-policy is conditioned only on the state; that is, it takes the form $\omega(s)$. In this case, it reduces to the formulation used in the original Option Keyboard paper; see Section 2.4 and Eq. (4) for details.
>
> To train $\omega(s)$, we use an actor-critic policy gradient approach, following the methodology of Barreto et al. (2019). Specifically, we follow Algorithm 3 (Appendix D), replacing $\phi$ with the scalar non-linear reward $r$, and $\psi^{\omega}(s, a)$ with a standard scalar Q-function $q^{\omega}(s, a)$. We will clarify these points in the updated version of the paper.
>
> ---
>
> ### Minor Typo
>
> Thank you for pointing out this minor typo. The correct notation is indeed $A^{w^*}_{r}(s,a)$ (i.e., $r$ subscripted), as we defined in Eq. (9). This will be fixed in the updated version of the paper.
>
> ---
>
> We hope our responses address the main point you raised regarding the computational cost of the method, as well as the other minor clarification questions. We will revise the text to incorporate a discussion about all these points. If you believe the clarifications above address the key points you raised, we'd be grateful if you would consider revisiting your score. If any points remain unclear, we welcome further feedback and would be happy to continue the discussion. Thank you again for your thoughtful comments!

---

### Official Review · Reviewer_1Yjh · 2025-07-02

**Clarity:** 4
**Significance:** 2
**Originality:** 3
**Rating:** 5
**Confidence:** 3

**Summary:**

The paper proposes a novel multi-task reinforcement learning method that uses a bi-level policy structure. In the linear-reward setting, prior work has shown that it is possible to compute a Convex Coverage Set (CCS), a set of base policies where the optimal policy for any reward function (by the linear reward definition) can be expressed as a convex combination of policies in the CCS. However, constructing a CCS can be expensive and it is hard to scale it to more complex domains. The proposed method additionally uses a meta policy that is allowed to dynamically choose the convex combination weights of the base policies depending on the state (unlike in prior work the weights are constant across states). This allows them to express policies that are optimal for more complex reward functions beyond the linear assumption. The proposed method is evaluated on four simulated environments. Compared to prior methods, the proposed method needs fewer number of base policies to achieve the same performance of the baselines with more base policies.

**Questions:**

N/A

**Ethical Concerns:**

["NO or VERY MINOR ethics concerns only"]

**Final Justification:**

The authors have addressed all of my concerns. The new experiments provide evidence that the proposed method OKB can scale to more complex domains. I am raising my score to 5.

**Limitations:**

yes

**Quality:**

3

**Strengths And Weaknesses:**

*Strengths*

- The paper is well-written and easy to read.
- The proposed method is novel and theoretically sound.

*Weaknesses*

- The authors consider only a limited number of baselines (SIP and DQN on one environment and SFOLS on others). The paper can benefit from including more baselines to demonstrate the effectiveness of the proposed method much better. For example, how does the proposed method compare to using existing unsupervised skill discovery methods that learn a latent-conditioned policies? (e.g., [1]). Also, comparing methods in options framework would also be relevant.
- One of the main motivations of the paper is that CCS "are computationally expensive and do not scale to complex domains". However, as far as I could tell. All the tasks considered are relatively simple where their CCS can be computed exactly. It would greatly strengthen the paper if the authors could showcase the propose method on some of the environments where computing a CCS is completely intractable.


[1] Benjamin Eysenbach, Abhishek Gupta, Julian Ibarz, and Sergey Levine. "Diversity is all you need: Learning skills without a reward function." arXiv preprint arXiv:1802.06070 (2018).

---

> ### Author Rebuttal · Authors · 2025-07-31
>
> We thank the reviewer for the positive comments regarding our method’s novelty and theoretical soundness, as well as the clarity and quality of the writing. The reviewer’s questions focused on two key aspects: **(1)** the set of baseline methods; and **(2)** the complexity of the domains used in the experiments. Below, we address both points.
>
> ---
>
> ### Comparison with Unsupervised Skill Discovery and Options Baselines
>
> We thank the reviewer for bringing up this important point. As the reviewer correctly noted, other methods from the unsupervised skill discovery and options literature are indeed relevant to the broader goal of more rapidly solving novel tasks. Examples include *Diversity is All You Need* (DIAYN), which learns a diverse set of skills without access to task rewards; *Set-Max Policy* (SMP), which constructs a set of policies that maximize worst-case performance over a task set; and *DSP*, which encourages diversity in the space of successor features while maintaining near-optimality (Zahavy et al., 2021).
>
> In Section 5, we chose SFOLS and SIP as our primary baselines, as both have been repeatedly shown in the literature to outperform more task-agnostic methods across a variety of settings. Moreover, they are representative of approaches that aim to construct policy sets with theoretical guarantees—a central objective of our work.
>
> Below, we present results comparing SIP to the three methods discussed above (DIAYN, SMP, and DSP), which belong to the family of approaches highlighted by the reviewer. These results are adapted from Alver and Precup (2022) and show that SIP consistently achieves better downstream task coverage (measured by mean normalized return over seventeen tasks uniformly covering the task space) in an experimental setting similar to ours. Since our results demonstrate that OKB systematically outperforms SIP—and SIP is well known to consistently outperform the other baselines in this class—these findings further support our choice of SIP as the most competitive and informative point of comparison.
>
> |  w  |  SIP  |  DIAYN  |  SMP  |  DSP  |
> |:---:|:-----:|:-------:|:-----:|:-----:|
> | 1   | 0.81  | 0.19    | 0.47  | 0.37  |
> | 2   | 0.86  | 0.23    | 0.50  | 0.39  |
> | 3   | 0.87  | 0.31    | 0.51  | 0.43  |
> | 4   | 0.87  | 0.44    | 0.54  | 0.50  |
> | 5   | 0.90  | 0.67    | 0.60  | 0.56  |
> | 6   | 0.90  | 0.84    | 0.62  | 0.67  |
> | 7   | 0.89  | 0.89    | 0.67  | 0.68  |
> | 8   | 0.91  | 0.91    | 0.68  | 0.68  |
> | 9   | 0.93  | 0.93    | 0.70  | 0.69  |
> | 10  | 0.94  | 0.94    | 0.73  | 0.71  |
> | 11  | 0.93  | 0.93    | 0.70  | 0.69  |
> | 12  | 0.95  | 0.90    | 0.67  | 0.66  |
> | 13  | 0.92  | 0.77    | 0.60  | 0.69  |
> | 14  | 0.94  | 0.65    | 0.58  | 0.51  |
> | 15  | 0.94  | 0.53    | 0.56  | 0.47  |
> | 16  | 0.93  | 0.38    | 0.54  | 0.44  |
> | 17  | 0.86  | 0.22    | 0.49  | 0.37  |
> | **Mean norm. return →**   | **0.90** | **0.63** | **0.60** | **0.56** |
> | **StdDev norm. return →** | **0.0388** | **0.2857** | **0.0832** | **0.1304** |
>
> *Table 1: Normalized returns (mean and standard deviation shown at the bottom) for various baselines, including SIP, DIAYN, SMP, and DSP across different weight vectors (tasks).*
>
> For completeness, we will include these comparisons in the paper. The findings above—namely, that SIP consistently outperforms more task-agnostic baselines such as DIAYN, DSP, and SMP in settings like the one we study—have been repeatedly observed in other prior work (e.g., Alegre et al., 2022). Based on the broader literature comparing these techniques, we thus fully expect to observe the same trend here, further reinforcing that SIP is the strongest and most informative point of comparison. All supporting tables and graphs will be added to Section 5 and the appendix.
>
> ---
>
> ### Complexity of the Tasks
>
> The second point raised by the reviewer concerns whether the domains used in our experiments are too simple and whether their CCS could be computed exactly.
> We have empirically evaluated OKB in challenging high-dimensional RL problems, including Minecart, Fetch-PickAndPlace (a robotic arm task), Item Collection, and Highway (an autonomous driving environment). These domains are well-established benchmarks in multi-task and multi-objective RL. Our results show that OKB consistently outperforms state-of-the-art GPI-based approaches, with its performance advantage becoming increasingly pronounced as task complexity (measured by the number of reward features $d$) increases. These results demonstrate that OKB is indeed applicable and highly effective in complex and challenging scenarios relevant to real-world applications, such as robotics and autonomous driving, going beyond simple simulated environments.
>
> Regarding tractability, prior to our paper, to the best of our knowledge, existing methods had been evaluated in problems with dimensionality at most $d=4$ (e.g., SFOLS and SIP). In our paper, we double the number of reward features and show that our method remains effective—outperforming all baselines even on problems an order of magnitude more challenging.
>
> Finally, we highlight that the CCS of some domains (e.g., FetchPickAndPlace and Highway) *cannot be computed exactly*. In all domains, we resorted to function approximation via neural networks to handle continuous state spaces of up to 25 dimensions. We agree with the reviewer on the importance of further discussing these points, and we will update the text to more clearly emphasize that our experiments were conducted on domains whose size and complexity had not been previously addressed in the literature.
>
> ---
>
> We hope our responses address your main questions. We will revise the text to incorporate a discussion on all the points brought up by the reviewer. If you believe the clarifications above address the key points you raised, we would be grateful if you would consider revisiting your score. If any points remain unclear, we welcome further feedback and would be happy to continue the discussion. Thank you again for your thoughtful comments!

---

> > ### Comment · Reviewer_1Yjh · 2025-08-01
> >
> > Thanks authors for the rebuttals. I appreciate the new comparisons to DIAYN and the clarification. Most of my concerns have been addressed except my concern on the task complexity. While it is nice to know that the CCS of some of the domains (e.g., FetchPickAndPlace and Highway) cannot be computed exactly, I am still not very confident that the proposed method can scale to more complex/challenging environments such as environments with image observations (e.g., MO-Supermario in the MO-Gym that the authors consider). While I remain positive about the work, I would like to maintain my current score.

---

### Official Review · Reviewer_rDD6 · 2025-07-06

**Clarity:** 3
**Significance:** 3
**Originality:** 4
**Rating:** 5
**Confidence:** 4

**Summary:**

This paper formalizes the idea of an optimal behavior basis for use with the Option Keyboard (OK), i.e., a set of base policies which are sufficient for OK to solve a family of MDPs with optimality. Further, the authors propose Option Keyboard Basis (OKB), an incremental method for contructing a behavior basis and meta-policy which combined are provably optimal with respect to any of the tasks in the family. The paper includes the results of several experiments intended to answer specific research questions and demonstrate the effectiveness of OKB compared to baselines in different domains.

**Questions:**

* Can you discuss how accurate the OK optimality test is in practice? Including results from domains where this can be checked exactly but are still large would be helpful.

* What other approaches did you consider for the OK optimality test and why did you choose the one that you did?

* Does using the OK framework specifically more constraining than a CCS? In other words, does this limit the problems OKB can be used to solve compared to previous methods?

**Ethical Concerns:**

["NO or VERY MINOR ethics concerns only"]

**Final Justification:**

The authors addressed the questions that I had regarding OKB. These were all things that I would like to see discussed somewhere in the paper, so I am happy that they intend to adjust their text to include them.

My rating remains Accept.

**Limitations:**

* The authors admit that the OK optimality test must be approximated, but do not expand much on how their approximation method was chosen or how much suboptimality it might introduce.

**Quality:**

3

**Strengths And Weaknesses:**

# Strengths
* The issue of discovery of a set of policies to use with Successor Feature-based methods which are sufficient to behave optimally in all tasks in a family is interesting and an important one to solve.
* The formalization of the optimal behavior basis is simple but still meaningful.
* OKB appears to be a strong approach to producing a set of base policies and meta-policy which are optimal with respect to the family of tasks while requiring fewer base policies than existing methods.
* The experiments are nicely targeted to answer the specific research questions that are proposed in Section 5.

# Weaknesses
* Only a single baseline that is not an ablation of OKB is used in each experiment. While this is sufficient to answer the specific research questions that the authors presented, it would be nice to see a broader comparison to compare this method to others that don't necessarily guarantee optimality across the family.
* For guaranteed optimality, the check for OK optimality requires knowledge of (or solving for) the optimal policy for the task in question, which is obviously not practical. While the sample-based method used in the experiments appears to be sufficient to allow OKB to be effective, there is little discussion of the best way to do this and no way to know how much suboptimality this approximation can introduce.

---

> ### Author Rebuttal · Authors · 2025-07-31
>
> We thank the reviewer for their positive assessment of our paper and for their encouraging comments on the relevance of the problem, the strength of our method, and the quality of our experiments. Below, we address the reviewer’s questions:
>
> ---
>
> ### Comparison with methods that do not guarantee optimality
>
> The reviewer highlighted that our comparisons were sufficient to positively answer all specific research questions and wondered how our method would perform when compared to weaker methods that do not necessarily guarantee optimality.
>
> We appreciate this question and agree that broader comparisons are valuable. For a more detailed discussion—including additional baselines from the unsupervised skill discovery literature—please see our response to Reviewer 1Yjh. Please do not hesitate to let us know if you have any thoughts or further questions based on those results: we would be happy to discuss them further if helpful.
>
> ---
>
> ### Do approximations to the OK optimality test affect correctness or optimality guarantees?
>
> Thank you for raising this important point. Your question helped us recognize the need to further clarify that our method does *not* rely on an exact test to guarantee the correctness of the behavior basis it constructs. As briefly noted in line 226 of Section 4.1, the test introduced in Eq. (9) for assessing OK optimality is a **sufficient (but *not* necessary) condition** for guaranteeing optimality. This test is designed to serve two key purposes: **(1)** to avoid training on tasks $w$ whose optimal policies have either already been learned or that can be reconstructed from $\Pi_k$ and $\omega$, and **(2)** to prioritize which tasks to train on in order to accelerate learning (see Appendix E for further discussion). In other words, the test is primarily a tool to reduce training effort, not a prerequisite for correctness.
>
> More concretely, even if the test is imperfect, as long as the algorithm avoids re-solving previously trained tasks, the algorithm is still guaranteed to identify a behavior basis that supports zero-shot optimal synthesis for all tasks in the family. In cases where the test is imperfect, the algorithm may take longer to converge, potentially solving more tasks than strictly necessary, but all correctness guarantees remain intact. These guarantees follow from our argument in lines 189–200 and from established results in the literature (e.g., Theorem 3 in Roijers, 2016). In short, although approximate versions of Eq. (9) may slightly increase the number of training iterations required, they do not affect the method’s correctness or result in suboptimal solutions.
>
> We appreciate your suggestion to make this point more explicit. In the final version, we will revise Section 4.1 to more clearly state that this test is a sufficient, but not necessary, condition for optimality.  We will also move portions of the discussion from Appendix E into the main text to better emphasize that its primary role is to accelerate the identification of a behavior basis.
>
> ---
>
> ### Is the OK framework more constraining than a CCS or prior methods?
>
> This is a great question. The set of problems solvable by the OKB is no more limited than that of previous approaches; in fact, it is *broader*. As shown, for example, by Alegre et al. (2022), a Coverage Convex Set (CCS) is guaranteed to contain the optimal solution to every and all tasks in the family defined in Eq. (5). Since our method can directly reconstruct any element of the CCS (and, importantly, without having to train an individual policy for each), it follows that it can optimally solve all linear problems in that family.
> This result is established formally in Theorem 4.3, where we prove that the OKB is at least as expressive as a CCS; that is, it can reconstruct all policies in the CCS, and potentially more. Proposition 4.4 goes further, showing that the OKB is in fact *strictly more expressive* than a CCS. In particular, it supports the direct reconstruction of optimal solutions not only for all linear tasks, but also for a relevant class of non-linear problems. This means that while existing methods that compute a CCS are limited to linearly expressible tasks, our method is not only guaranteed to return zero-shot optimal solutions to all such tasks, but it can also—importantly—solve a broader class of non-linear problems that previous approaches could not handle.
>
> ---
>
> We hope the comments above address your questions. Please feel free to reach out with any further thoughts—we would be glad to continue the discussion. Thank you once again for your thoughtful feedback and for taking the time to carefully read our work and share your insights.

---

> > ### Comment · Reviewer_rDD6 · 2025-08-06
> >
> > I thank the authors for their answers to my questions as they helped clarify some aspects of OKB in my mind. I will keep my Accept rating.

---

> > > ### Author Response · Authors · 2025-08-08
> > >
> > > We thank the reviewer once again for the thoughtful discussion, constructive feedback, and valuable comments. We greatly appreciate their positive and encouraging remarks on the relevance and importance of our contributions, as well as their support for the acceptance of our paper.

---

### Decision · Program_Chairs · 2025-09-17

**Decision:**

Accept (poster)

**Comment:**

The paper proposes a method to construction an option-keyboard basis, a basis of behaviors that can be used to express the solution for all MDPS within a family, within the options keyboard framework. The argument is centered around the idea of a CCS (convex coverage set), which is a set of policies that include an optimal policy for any linear task. The challenge with a CCS is that it often grows exponentially with the number of reward features. The proposed methodology incrementally constructs a basis of behavior that can express all policies in a CCS, but significantly more compactly. The authors provide empirical and theoretical results to this effect and validate on a number of simulated problems.

Reviewers brought up several key concerns during discussion - 1) limited baselines, 2) limited task complexity, 3) theoretical justification of complexity. Through the rebuttal period and discussion the authors were able to include more thorough comparisons to unsupervised RL methods, increase task complexity to other tasks like MO-supermario and provide an argument on the theoretical merits of their approach. The method does still need further validation and comparisons to make it really stand out in the literature, but it is a good addition to the community.